# Principles of gamma synchrony predict figure–ground perception in texture stimuli

Maryam Karimian[1,2,3,4], Mark Jonathan Roberts[1,5], Peter De Weerd[1,3,5†], Mario Senden[1,5*†]

[1]Department of Cognitive Neuroscience, Faculty of Psychology and Neuroscience Maastricht University, Maastricht, Netherlands; [2]Science of Intelligence, Research Cluster of Excellence, Berlin, Germany; [3]Maastricht Centre for Systems Biology (MaCSBio), Maastricht University, Maastricht, Netherlands; [4]Institute for Theoretical Biology, Department of Biology, Humboldt-Universität zu Berlin, Berlin, Germany; [5]Maastricht Brain Imaging Centre, Faculty of Psychology and Neuroscience, Maastricht University, Maastricht, Netherlands

## eLife Assessment

Karimian et al. present a **valuable** new model to explain how gamma-band synchrony (30-80 Hz) can support human visual feature binding by selectively grouping image elements, countering recent criticisms that the stimulus dependence of gamma oscillations limits their functional role. Grounded in the theory of weakly coupled oscillators the model captures behavioural patterns observed in human psychophysics, offering support for the potential role of synchrony-based mechanisms in feature-binding. The development of the model in alignment with primate electrophysiology **convincingly** supports the paper's claims that gamma synchrony may be the underlying mechanism. While the paper does not present electrophysiological results that directly link gamma oscillations to figure-ground segregation in the presented task, the model makes several predictions that can be tested experimentally.

**\*For correspondence:**
mario.senden@
maastrichtuniversity.nl

†These authors contributed
equally to this work.

**Competing interest:** The authors declare that no competing interests exist.

**Abstract** Gamma synchrony is ubiquitous in visual cortex, but whether it contributes to perceptual grouping remains contentious based on observations that gamma frequency is not consistent across stimulus features and that gamma synchrony depends on distances between image elements. These stimulus dependencies have been argued to challenge the idea that the visual system groups image elements by synchronizing the neural assemblies that encode them. Here, we argue instead that these dependencies may shape synchrony in perceptually meaningful ways. Indeed, according to the theory of weakly coupled oscillators (TWCO), synchrony-based grouping mechanisms require stimulus dependence. Synchronization among coupled oscillators depends on frequency dissimilarity and coupling strength, which in early visual cortex relate to local feature dissimilarity and physical distance, respectively. We manipulated these factors in a texture segregation experiment wherein human observers identified the orientation of a figure defined by reduced contrast heterogeneity compared to the background. Human performance followed TWCO predictions both qualitatively and quantitatively, as formalized in a computational model. Moreover, we found that when enriched with a Hebbian learning rule, our model also predicted human learning effects: Increases in model gamma synchrony due to perceptual learning predicted improvements in texture segregation across sessions. Taken together, our data suggest that the stimulus-dependence of gamma synchrony captures local image statistics and is linked to the stimulus-dependence of texture segregation,

and that the effect of visual experience on gamma synchrony provides a viable perceptual learning mechanism for training-induced improvements in texture segregation. Our results suggest that gamma synchrony with its inherent stimulus dependencies can provide a plausible mechanistic basis for perceptual grouping and visual scene segmentation.

## Introduction

Oscillations are ubiquitous in the cortex (*Buzsáki et al., 2013*) and can synchronize both within and between cortical areas (*Anand et al., 2023*; *Lowet et al., 2017*; *Melloni et al., 2007*), but whether this contributes to neural information processing remains a matter of debate (*Doelling and Assaneo, 2021*; *Duecker et al., 2021*; *Fernandez-Ruiz et al., 2023*; *Ray and Maunsell, 2015*; *Roelfsema, 2023*). Early suggestions that synchrony in the gamma frequency band (30–80 Hz) plays a central role in visual feature binding (*Singer, 1999*; *Uhlhaas et al., 2008*) have been called into question based on observations that the gamma frequency is not consistent across stimulus features (*Ray and Maunsell, 2010*; *Ray and Maunsell, 2015*; *Shirhatti et al., 2022*) and depends on distances between image elements (*Roelfsema, 2023*; *Roelfsema et al., 2004*), making it difficult to group components of the same object by synchrony among the neural assemblies encoding these components (*Dubey and Ray, 2020*; *Roelfsema, 2023*). Alternatively, it has been proposed that the stimulus dependence of gamma synchrony facilitates, rather than hinders, their functional significance for visual processing by allowing contiguous neural assemblies that share a sufficiently similar oscillation frequency to synchronize into meaningful groups, while also blocking synchrony among assemblies with substantial frequency difference or physical separation (*Lowet et al., 2015*; *Lowet et al., 2017*). Here we show empirical and computational support for this view.

Analyzing a visual scene requires integration of features into coherent objects (feature binding), but also segregation of features belonging to distinct objects (feature separation). It remains unclear how this is achieved, but the stimulus dependence of gamma may be critical for a synchrony-based neural grouping mechanism that achieves both feature binding and separation. This idea is rooted in the theory of weakly coupled oscillators (TWCO), which describes the preconditions for synchrony among coupled oscillators (*Acebrón et al., 2005*; *Ermentrout et al., 2019*; *Kuramoto, 1984*; *Neu, 1979*; *Strogatz, 2000*). A group of coupled oscillators synchronizes if the discrepancy in their frequencies, referred to as their detuning, is overcome by the strength of their connection, referred to as their coupling. Thus, synchrony can occur even in the presence of strong detuning, if the coupling strength is sufficiently high, whereas if the coupling strength is low, synchrony can only occur if the detuning is also minimal. This relationship can be graphically depicted in an Arnold tongue (*Coombes and Bressloff, 1999*; *Pikovsky et al., 2001*), which shows the regions where synchrony occurs based on the balance between detuning and coupling strength (see *Figure 1a* for an illustration). These abstract principles are concretely realized in early visual cortex. Neural assemblies exhibit gamma oscillations in their population activity at frequencies that are directly related to stimulus features such as spatial frequency, contrast, and orientation (*Dubey and Ray, 2020*; *Henrie and Shapley, 2005*; *Shapira et al., 2017*), and particularly contrast (*Hadjipapas et al., 2015*; *Lowet et al., 2015*; *Roberts et al., 2013*). In early visual cortical areas, coupling strength between neural assemblies is directly related to the efficacy of lateral anatomical connectivity, which declines with cortical distance (*Boucsein et al., 2011*; *Gilbert and Wiesel, 1983*; *Lowet et al., 2015*; *Lowet et al., 2017*; *Stettler et al., 2002*; *Ts'o et al., 1986*). In conjunction with the retinotopic organization of early visual cortex, this implies that neural assemblies encoding nearby visual regions are more strongly coupled. Taken together, synchrony in early visual cortex could occur across widely spaced neuronal assemblies in response to scenes with low feature heterogeneity, but only for closely spaced assemblies in response to scenes with high feature heterogeneity (see *Figure 1b* for an illustration). Indeed, a recent electrophysiological study in macaque V1 in which cortical distance and stimulus contrast heterogeneity were parametrically manipulated has confirmed that gamma synchrony behaves in line with the principles of TWCO (*Lowet et al., 2017*).

A synchrony-based grouping mechanism based on these principles has been successfully exploited for image segmentation in machine vision (*Fang et al., 2014*; *Lowet et al., 2015*; *Nikonov et al., 2020*). Here, we bring these perspectives together to test whether human vision likewise behaves in accordance with TWCO principles. To test this hypothesis, we used a figure-ground segregation

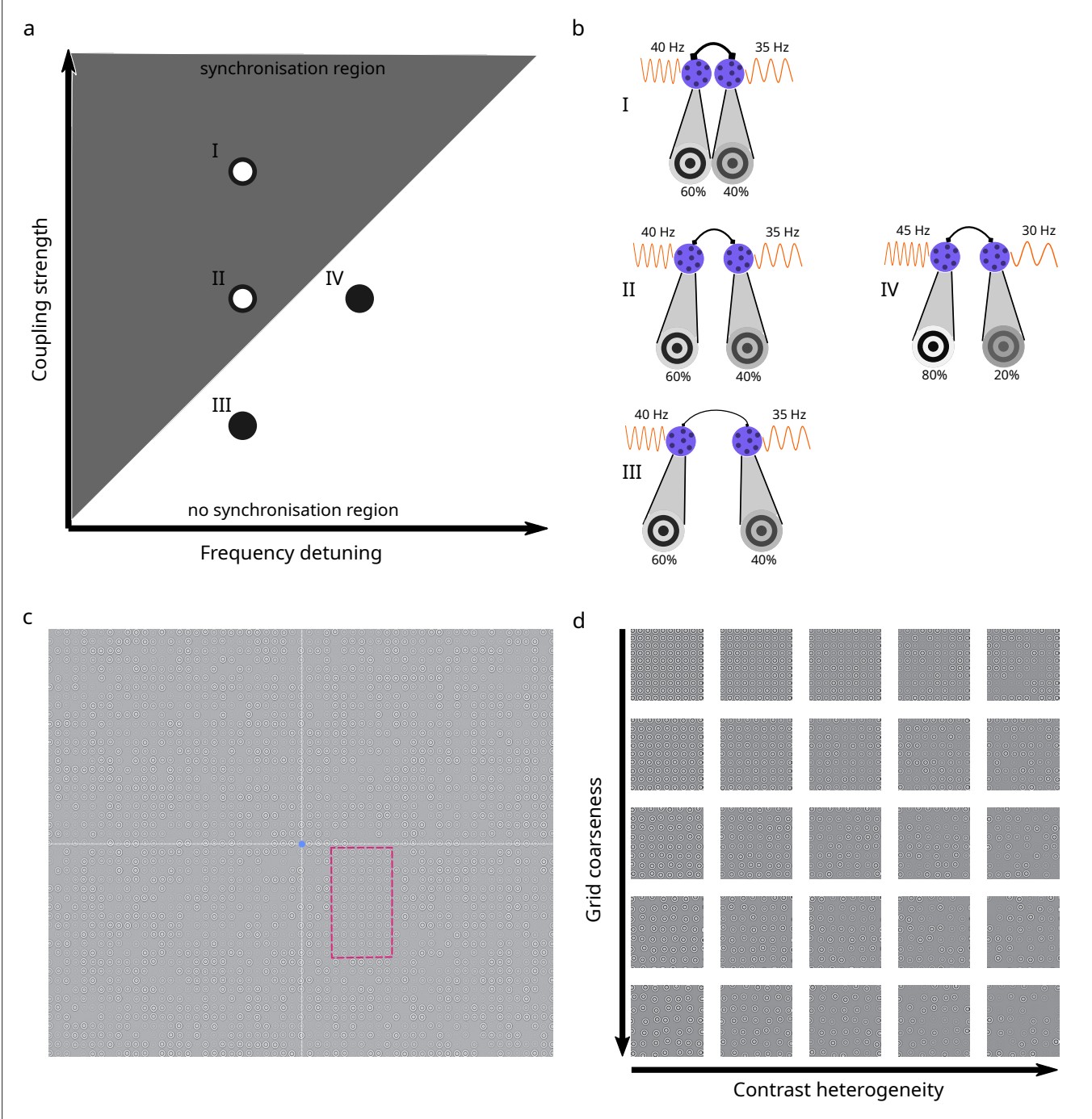

**Figure 1.** Schematic illustration of synchronization principles in visual cortex and stimulus design. (**a**) Arnold tongue: triangular region shows combinations of detuning and coupling strength that allow synchrony (light gray). Open circles indicate two scenarios conducive to synchrony. The first scenario (**I**) combines strong coupling with moderate detuning. The second scenario (**II**) combines moderate coupling with moderate detuning. Closed circles indicate two scenarios not conducive to synchrony. The third scenario (**III**) combines weak coupling with moderate detuning. The fourth scenario (**IV**) combines moderate coupling with large detuning. (**b**) Translation of the four scenarios to stimulus features. Detuning and coupling strength map onto contrast heterogeneity and grid coarseness through the anatomy and physiology of early visual cortex. In this simplified illustration, two texture elements (Gabor annuli) fall within receptive fields of neural assemblies (purple) in early visual cortex. Contrast determines oscillation frequency (orange), with higher contrasts leading to higher frequencies. Differences in contrast within the receptive fields of two neural assemblies thus lead to differences in their frequencies and hence to higher detuning. Note that neural assemblies typically have several Gabor annuli in their receptive fields and extract their average contrast. There is thus no one-to-one mapping between annuli and receptive fields in our model. Coupling strength (line thickness of connecting arrow) depends on cortical distance, which due to retinotopy directly relates to the distance between texture elements in the visual field.

*Figure 1 continued on next page*

*Figure 1 continued*

Larger distances between annuli thus stimulate more remote neural assemblies with weaker coupling. (c) Example full texture stimulus comprised of nonoverlapping Gabor annuli on irregular grid. For all participants and sessions 1–8, the lower right quadrant contains a vertical figure (magenta outline, not shown to participants). Blue dot: fixation point. Axes separating quadrants shown for illustration only, not visible to participants. On a given trial, the figure may be vertical or horizontal and participants indicated the figure's orientation. (d) Figure region cut-outs illustrating experimental conditions. Grid coarseness (five steps) manipulates coupling strength for both figure and background. Contrast heterogeneity (five steps) manipulates detuning within figure. Background always at maximum heterogeneity (equivalent to rightmost column). The 25 cut-outs show all combinations of grid coarseness and contrast heterogeneity used in the experiments.

paradigm wherein human observers reported the orientation of a rectangular figure region in a texture stimulus composed of Gabor annuli (see *Figure 1c* for an illustration). According to TWCO, synchrony is governed by the interplay between oscillator detuning and coupling strength (*Acebrón et al., 2005*). Therefore, we created stimuli in which these two core parameters were systematically manipulated. We parametrically varied contrast heterogeneity as an implementation of frequency detuning and grid coarseness as an implementation of the cortical distance that determines coupling strength (see *Figure 1d* for an illustration). The figure was defined by a less heterogeneous contrast distribution between the elements, compared to elements in the background. Additionally, we investigated whether this synchrony-based grouping mechanism is adaptive by using a perceptual learning paradigm in which participants improved their perceptual performance over eight daily sessions. By formalizing the principles of TWCO in a V1 oscillator model augmented with a simple Hebbian learning mechanism, we derived quantitative predictions from the theory. Our psychophysics results align well with the synchrony exhibited by the model, supporting the idea that stimulus-dependent gamma synchrony may be behaviorally relevant.

## Results

Eight participants (six female, mean age 23.75, standard deviation 6.4536) performed a two-alternative forced choice texture discrimination task. We employed a repeated-measures design with extensive sampling. A design analysis indicated that our sample size afforded approximately 92% posterior detection probability (analogous to statistical power) for the core effects (*Supplementary file 1*) and robustness to both type-S and type-M errors. Texture stimuli consisted of nonoverlapping Gabor annuli on an irregular grid (see *Figure 1c*). Each Gabor annulus was characterized by its own local contrast and was equiluminant with the background. Within a single visual quadrant, a rectangular figure was defined by less heterogeneity in the contrasts of local Gabor elements compared to the background, while keeping mean contrast between figure and background equal. Participants indicated the orientation (horizontal vs vertical) of the figure while fixating centrally. We manipulated two factors. The first was contrast heterogeneity within the figure, which we operationalized as the width of a uniform distribution from which annulus contrast values were drawn. This distribution was centered around a mean contrast of 50%. The background exhibited maximum contrast heterogeneity (from 0% to 100%). The second factor was the coarseness of the grid (distance between annuli). This manipulation affected figure and background equally. Both factors were manipulated in five steps resulting in 25 conditions (see *Figure 1d*). Within an experimental session, participants completed 30 blocks of each condition (750 trials). Participants received feedback after each trial in the form of color changes of the fixation point. Eye-tracking was used to ensure fixation, and trials where fixation was broken during either the fixation period preceding the stimulus, or during stimulus presentation, were aborted and repeated at a randomly chosen time later in the session. The experiment consisted of nine consecutive sessions (eight training and one transfer session). In the transfer session, the rectangular figure was moved to the diagonally opposite quadrant.

To provide a mechanistic link between contrast heterogeneity, grid coarseness, and synchrony in early visual cortex on the one hand, and quantitative predictions of discrimination accuracy on the other, we developed a phase-oscillator model of V1. The model represents a patch of visual space corresponding to the figure region in our psychophysics experiments, mapped onto V1 using a complex-logarithmic topographic transformation (*Balasubramanian and Schwartz, 2002*; *Schwartz, 1980*). To reduce computational cost, we only modeled the figure and not the background, under the assumption that the synchrony level in one image region would not be substantially altered by

the synchrony level in the other image region (see *Figure 2—figure supplement 1*). Based on this, synchrony in the background at maximum contrast disparity was equated to synchrony in the figure at that contrast disparity. Each model oscillator represents a neural assembly receiving local input from the visual field. The frequency of each oscillator is a quasi-linear function of the contrast falling inside its receptive field, as has been determined previously in macaques (*Evers et al., 2021*; *Roberts et al., 2013*). Receptive fields are modeled as isotropic 2D Gaussian functions with sizes that scale with eccentricity according to human cortical magnification (*Freeman and Simoncelli, 2011*). Further-more, we included recurrent connections between phase oscillators reflecting the lateral anatom-ical connectivity among columns in V1 and other low-level visual areas (*Crist et al., 2001*). In line with anatomical data (*Amir et al., 1993*; *Eckhorn, 1994*; *Gilbert and Wiesel, 1989*; *Ts'o et al., 1986*), coupling strength in our model declines exponentially with physical distance along the cortical surface. Our model captures this with two parameters estimated from independent neurophysiolog-ical data (*Lowet et al., 2017*): maximum coupling strength γ and coupling decay factor $\lambda$. The model was exposed to the same figure region texture stimuli as human participants, with manipulations of contrast heterogeneity and grid coarseness. We quantified the model's degree of zero-lag synchrony as the magnitude of the Kuramoto order parameter (synchronization index).

In our V1 model, learning is implemented to occur offline between simulated sessions, following a Hebbian-type learning rule that adapts coupling strengths based on pairwise phase-locking values (PLVs) accumulated over trials within a session. The contribution of each trial to learning is weighted by the probability of a correct response, determined by a psychometric function relating model synchrony to performance. This learning mechanism implies that connections between oscillators that exhibited coherence on correct trials are strengthened, bounded by the maximum coupling strength. Incorpo-rating an upper bound on connections was motivated by findings that synaptic strength is limited by intrinsic properties of vesicular docking (*Malagon et al., 2020*) and that late long-term potentiation approaches a maximum after several repeated experiences (*Kandel et al., 2000*). Free parameters of the learning mechanism were estimated using data from the first two experimental sessions. To maxi-mally disentangle data used for adjusting model parameters and data used for testing model predic-tions, we employed a leave-one-out cross-validation procedure. Model parameters were repeatedly estimated from the first two sessions in seven of our eight participants, and the resulting model was used to predict performance in the remaining six sessions of the left-out participant. Our model rests on the assumption that learning-induced structural changes in early visual cortex are specific to the retinotopic locations of the trained stimuli. We evaluated whether this assumption holds for our human participants using the transfer session following the main training period. In the transfer session, participants performed the texture discrimination task with the figure region moved to a visual quadrant that had not been previously exposed to the figure. If learning is indeed local, partic-ipants' performance in the transfer session should resemble that of early training sessions, indicating a reset in performance for the new retinal location. On the other hand, if learning generalizes across retinal locations, performance in the transfer session should maintain the improvements seen in later training sessions. By comparing transfer session performance to both early and late training sessions, we can evaluate the validity of our model's assumption.

## Synchrony principles govern static figure-ground perception

We first asked the question whether the factors that determine synchrony among coupled oscillators, frequency detuning, and coupling strength are predictive of the human ability to segregate a rectan-gular figure from its background in texture stimuli. In early visual cortex, oscillation frequency directly maps onto the contrast of texture elements (*Hadjipapas et al., 2015*; *Lowet et al., 2015*; *Roberts et al., 2013*) and coupling strength directly maps onto their physical proximity (*Gilbert and Wiesel, 1983*; *Lowet et al., 2015*; *Lowet et al., 2017*; *Stettler et al., 2002*; *Ts'o et al., 1986*). If texture segregation indeed depends on the synchrony principles identified by the theory of weakly coupled oscillators (TWCO), we expect discrimination accuracy to reveal a 'behavioral' Arnold tongue in the space defined by contrast heterogeneity and grid coarseness.

To test these predictions, we analyzed the main effects of contrast heterogeneity and grid coarse-ness, as well as their interaction, on discrimination accuracy using Bayesian hierarchical logistic regression. This allowed us to analyze individual trial data rather than aggregated accuracy, while simultaneously accounting for within-subject variability by estimating participant-specific intercepts

and slopes for each predictor. Both contrast heterogeneity and grid coarseness were z-normalized prior to fitting the statistical model. Note that while the principles of TWCO primarily predict main effects of contrast heterogeneity and grid coarseness, we additionally included their interaction to capture complex relationships specific to V1 that are not immediately apparent from the general theory. Specifically, coupling strength decays exponentially with cortical distance, which itself depends on cortical magnification. This should lead to a highly nonlinear relationship between grid coarseness and coupling strength that is likely to manifest as an interaction. In line with our expectations, the test provided strong evidence that both increased contrast heterogeneity ($\beta = -0.60$, 95% HDI [−0.89,−0.30], Pr[$\beta<0$]=0.999, OR = 0.56, 95% HDI for OR [0.41, 0.74]) and grid coarseness ($\beta = -0.27$, 95% HDI [−0,40–0.13], Pr[$\beta<0$]=0.999, OR = 0.77, 95% HDI for OR [0.67, 0.88]) reduced discrimination accuracy (*Figure 2a and b*). These results provide credible evidence that a one-standard-deviation increase in contrast heterogeneity reduces the odds of a correct response by approximately 44%, while a similar increase in grid coarseness reduces the odds by 23%. Furthermore, *Figure 2a and b* show a behavioral Arnold tongue as a triangular region of high accuracy (≥75% correct). There was likewise strong evidence for an interaction between contrast heterogeneity and grid coarseness ($\beta$=0.24, 95% HDI [0.12, 0.36], Pr[$\beta>0$]=0.998, OR = 1.27, 95% HDI for OR [1.12, 1.44]). This indicates that the specific characteristics of early visual cortex contribute beyond the general principles of TWCO.

Our model of V1 captures both the general principles of TWCO as well as idiosyncratic characteristics of early visual cortex in a single mechanism, and we expected this model to predict the human ability to segregate a rectangular figure from its background in texture stimuli. Indeed, the synchrony exhibited by our model (*Figure 2c and d*), when exposed to the same stimuli as our participants, resembled behavioral discrimination accuracy (*Figure 2a and b*). A Bayesian hierarchical logistic regression with model synchrony as the sole predictor revealed strong evidence that it is associated with improved accuracy ($\beta$=0.76, 95% HDI [0.33, 1.21], Pr[$\beta>0$]=0.998, OR = 2.19, 95% HDI for OR [1.38, 3.33]). This represents credible evidence that a one-standard-deviation increase in synchrony more than doubles the odds of a correct response. Hence, our proposed mechanism is capable of reproducing the key patterns in the behavioral data.

A natural question is whether synchrony constitutes the unique mechanistic link from stimulus features to perception within our model. To address this question, additional analyses used the average model firing rates within the figure as a predictor for segregation, as well as the difference between average model firing rates inside and outside the figure. The latter rate difference between figure and ground can serve as a phenomenological proxy for putative rate-based segregation mechanisms. Note that we treat the instantaneous frequency of each oscillator as a proxy for the instantaneous population firing rate of the corresponding neural assembly.

With respect to the average figure firing rates, we found some evidence indicating that they were associated with segregation accuracy ($\beta$=0.07, 95% HDI [–0.025, 0.16], Pr[$\beta>0$]=0.941, OR = 1.07, 95% HDI for OR [0.98, 1.18]). However, because the 95% highest density interval included zero, we evaluated whether the effect fell within a Region Of Practical Equivalence (ROPE) of ±2% accuracy and found only weak evidence for this (Pr[|$\Delta$acc|<0.02]=0.680). Hence, the effect is likely present, but small. With respect to rate differences, we found credible evidence that they could be associated with accuracy ($\beta$=–0.55, 95% HDI [-0.78,–0.32], Pr[$\beta<0$]=0.999, OR = 0.58, 95% HDI for OR [0.46, 0.73]). However, the effect of rate difference was smaller than that of synchrony. Furthermore, firing in the figure was reduced compared to background firing.

We next compared synchrony, average figure firing rate, and rate differences derived from the same V1 simulations in terms of their out-of-sample predictive accuracy using Pareto-smoothed importance sampling leave-one-out cross-validation. Synchrony was favored over rate difference ($\Delta$ELPD ≈ 19, dSE ≈ 14) and average figure firing ($\Delta$ELPD ≈ 127, dSE ≈ 17). The stacking weights further support this with synchrony receiving a weight of 0.90, rate difference a weight of 0.10, and the average figure firing rates a weight of effectively zero. These results indicate that, for our stimuli and our V1 model, a synchrony-based readout provides the most faithful mapping from stimulus to perception among simple alternatives. However, this comparison does not rule out that more sophisticated rate-based models could provide viable mechanistic accounts of figure-ground segregation. Nevertheless, our data indicate that synchrony-based mechanisms are eminently viable.

A key strength of our model is that it does not depend on fine-tuning parameters to our behavioral data. To demonstrate this, we conducted a parameter space exploration of key choices of

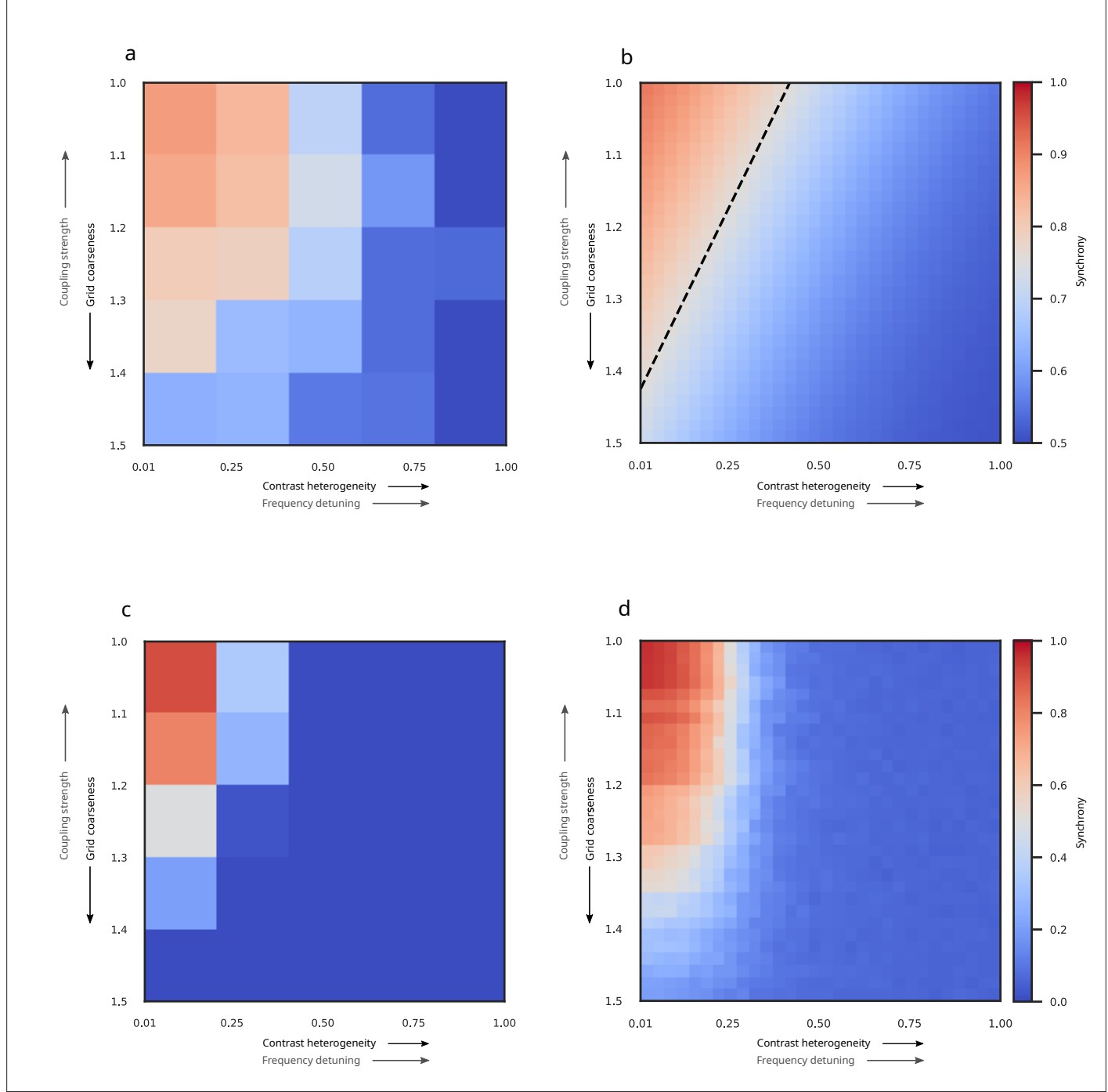

**Figure 2.** Behavioral and simulated Arnold tongues. (**a**) Average discrimination accuracy for each of the 25 experimental conditions revealed a behavioral Arnold tongue in the space defined by contrast heterogeneity and grid coarseness. Contrast heterogeneity translates into the variance of frequencies (detuning), whereas grid coarseness translates into cortical distance (coupling strength). (**b**) Fitted behavioral Arnold tongue after fitting a two-dimensional psychometric curve to the results in (**a**). The dashed line indicates the combination of contrast heterogeneity and grid coarseness corresponding to 75% accuracy. (**c**) Zero-lag synchrony among model oscillators showing an Arnold tongue in the same parameter space as (**a**). Simulation conditions matched the 25 experimental conditions. (**d**) High-resolution visualization of zero-lag synchrony, using 900 conditions (30 levels each of contrast heterogeneity and grid coarseness) to provide a more detailed representation of the Arnold tongue.

The online version of this article includes the following figure supplement(s) for figure 2:

**Figure supplement 1.** Model-derived quantities for different combinations of maximum coupling and decay rate.

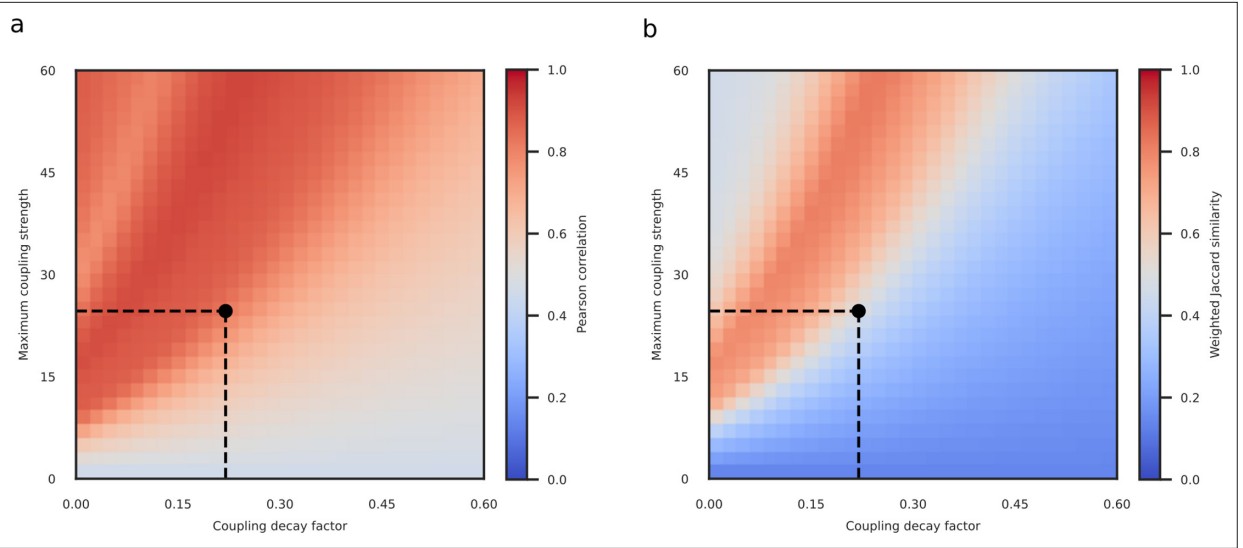

**Figure 3.** Comparison of behavioral and simulated Arnold tongues across coupling parameter space. (**a**) Pearson correlation between the behavioral Arnold tongue and simulated Arnold tongues obtained from models with coupling weights determined by different combinations of maximum coupling strength and coupling decay factor. The point labeled by the black circle shows the combination of parameters that were obtained from independent (macaque) data. (**b**) Weighted Jaccard similarity between the behavioral Arnold tongue and simulated Arnold tongues. This metric is displayed across the same parameter space as in (**a**).

model parameter values (maximum coupling strength and coupling decay factor) and found that our choices, which were obtained from independent observations in macaques (*Lowet et al., 2017*), were already close to optimal. We used Pearson correlations (*Figure 3a*) and weighted Jaccard similarity (*Figure 3b*) to assess the similarity between the behavioral Arnold tongue and the Arnold tongue predicted by our V1 model for various combinations of maximum coupling strength and coupling decay factor. We included both correlations and Jaccard similarity because the former is more widely known while the latter is more conservative. To compute weighted Jaccard similarity between two sets of real numbers, they need to fall within the same range. Accordingly, we applied min-max normalization to ensure that discrimination accuracy fell within a zero-to-one range matching the range of the synchronization index. This procedure yielded the similarity comparisons color-coded in *Figure 3*. The point marked with the black dot reflects the parameter value combination that was based on independent macaque data (24.63 and 0.22, respectively) and that was exclusively used for our model predictions. When using Pearson correlations as a similarity measure, this parameter value combination fell just within the region of optimal parameter values for our behavioral results (*Figure 3a*). When using the more conservative weighted Jaccard similarity index (*Figure 3b*), our chosen parameter value combination appeared slightly outside of the optimal region. Thus, the two model parameters estimated from neurophysiological recordings in monkeys were close to optimal for predicting human perceptual behavior, but not fully optimal. This may be due to horizontal connections in human visual cortex extending further than those in the macaque (*Amir et al., 1993*; *Burkhalter and Bernardo, 1989*; *Lund et al., 1993*; *Voges et al., 2010*; *Yoshioka et al., 1996*), suggesting a slightly smaller coupling decay factor in humans. Applying a smaller coupling decay would move model parameters into the optimal regime, thereby extending the predicted Arnold tongue (*Figure 2c*) diagonally in the direction of the behavioral Arnold tongue (*Figure 2a*). The parameters maximum coupling and decay factor might reflect biological constraints on the strength of lateral connections (*Kandel et al., 2000*; *Malagon et al., 2020*; *Rioult-Pedotti et al., 1998*), which to some extent may differ between monkeys and humans.

## Plasticity-induced changes in synchrony quantitatively predict perceptual learning

Our results show that a neural grouping mechanism based on synchrony principles can account for behavioral performance, suggesting it is a viable candidate for explaining texture segregation. We

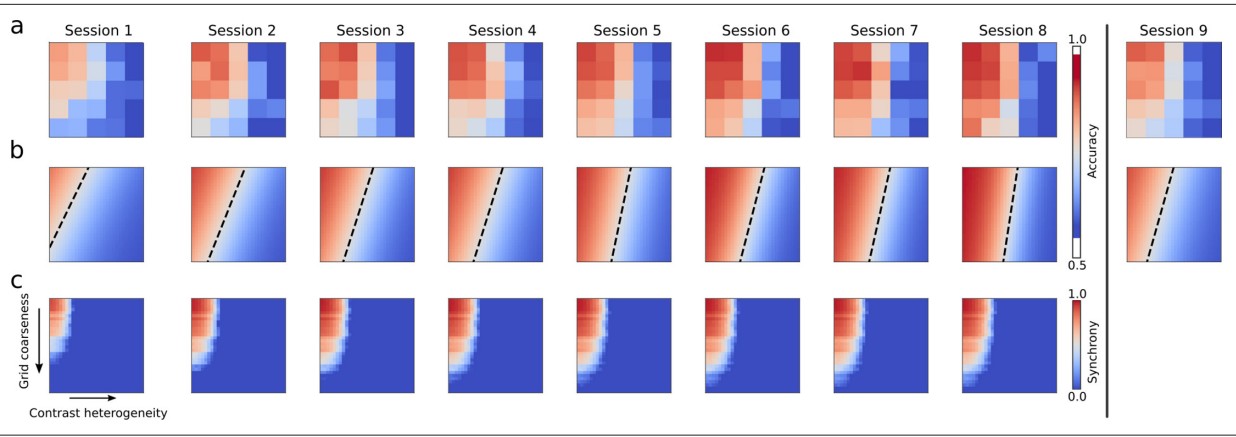

**Figure 4.** Learning effects on Arnold tongues. (**a**) Group average behavioral Arnold tongues for the 25 experimental conditions for each session. The vertical black line separates transfer session 9 from training sessions 1–8. (**b**) Two-dimensional psychometric curves fitted to session-specific group average behavioral Arnold tongues. The dashed line again indicates the combination of contrast heterogeneity and grid coarseness at which participants achieve 75% accuracy. (**c**) Simulated Arnold tongues for each of the eight training sessions including session-by-session learning in the model. We did not include a simulation of the ninth session because the location-specificity of the model learning rule would render it identical to the first session. Note that for visualization purposes, we simulated the model for 30 levels of contrast heterogeneity and 30 levels of grid coarseness, in both cases, including the five levels investigated experimentally.

The online version of this article includes the following figure supplement(s) for figure 4:

**Figure supplement 1.** Synchronization behavior in stimuli with figure and background regions.

next asked whether training-induced changes of lateral connections among neural assemblies in early visual cortex affect assemblies' readiness to synchronize and whether this is accompanied by performance improvements. We reasoned that neural synchrony must remain adaptable to the statistics of visual experiences to function effectively as a grouping mechanism. Consequently, we hypothesized that if synchrony among neural assemblies is related to figure-ground segregation and enhanced through perceptual learning, the ability to segregate figure from ground should increase with training.

To test this, both the model and human participants were exposed to eight daily sessions of extensive training using identical stimuli and experimental conditions. We hypothesized that both grid coarseness and contrast heterogeneity exhibit main effects on discrimination accuracy. However, we also expected coupling strength to increase with learning and that this would allow synchrony to occur for increasingly coarser grids. We thus hypothesized an interaction effect between session and grid coarseness on discrimination accuracy. Furthermore, we hypothesized an additional interaction between session and contrast heterogeneity where the effect of contrast heterogeneity would increase over sessions. Model simulations for the first session never revealed a synchronized state for contrast heterogeneity values beyond 0.25, even for the densest grids (see *Figure 2c and d*). This, together with an upper bound on coupling strength, suggested that synchrony cannot be achieved far beyond this cutoff point, even after extensive training. Indeed, model simulations of training confirmed this, showing that synchrony approached this cutoff point for increasingly coarser grids over sessions (*Figure 4c*). These model results indicate that the effect of contrast heterogeneity would increase over sessions with high performance for values below the cutoff point and low performance above the cutoff point. Finally, extensive training may globally increase participants' performance, implying a main effect of session.

To evaluate these predictions, we performed Bayesian hierarchical logistic regression including the main effects of contrast heterogeneity, grid coarseness, and session, as well as interactions between session and contrast heterogeneity, between session and grid coarseness, and between contrast heterogeneity and grid coarseness. As before, we included the interaction between contrast heterogeneity and grid coarseness to account for potential nonlinear effects specific to V1. The analysis revealed strong evidence that participants' ability to segregate figure from ground increased over sessions ($\beta$=0.095, 95% HDI [0.066, 0.126], Pr[$\beta$>0]=1.00, OR = 1.10, 95% HDI for OR = [1.07, 1.13]). Furthermore, we found strong evidence for an interaction between contrast heterogeneity and session ($\beta$ = −0.081, 95% HDI [−0.090,−0.070], Pr[$\beta$<0]=1.00, OR = 0.92, 95% HDI for OR = [0.91, 0.93]). We

**Table 1.** Effects of contrast heterogeneity on discrimination accuracy across sessions.

HDI = Highest Density Interval. Log-odds and Odds Ratios represent the effect of a one standard deviation increase in contrast heterogeneity on the odds of correct discrimination. The probabilities of negative log-odds are for the simple effect of contrast heterogeneity in each session.

| Session | Log-odds ($\beta$) | Odds ratio (OR) | 95% HDI for OR | Pr[$\beta$<0] |
|---|---|---|---|---|
| 1 | −0.66 | 0.52 | [0.38, 0.66] | 1.00 |
| 2 | −0.74 | 0.48 | [0.35, 0.61] | 1.00 |
| 3 | −0.82 | 0.44 | [0.32, 0.56] | 1.00 |
| 4 | −0.90 | 0.41 | [0.30, 0.52] | 1.00 |
| 5 | −0.98 | 0.38 | [0.28, 0.48] | 1.00 |
| 6 | −1.06 | 0.34 | [0.26, 0.44] | 1.00 |
| 7 | −1.14 | 0.32 | [0.24, 0.41] | 1.00 |
| 8 | −1.22 | 0.30 | [0.22, 0.38] | 1.00 |

also found evidence for a small interaction between grid coarseness and session ($\beta$ = −0.008, 95% HDI [−0.018, 0.001], Pr[$\beta$<0]=0.954, OR = 0.99, 95% HDI for OR = [0.98, 1.00]). However, the 95% highest density interval included zero. We subsequently confirmed that the change was within a region of practical equivalence (ROPE) of ±2% accuracy Pr[|Δacc|<0.02]=0.999. While the interaction between session and grid coarseness is thus negligible, there was strong evidence for a main effect of grid coarseness on discrimination accuracy ($\beta$ = −0.316, 95% HDI [−0.377,−0.262], Pr[$\beta$<0]=1.00, OR = 0.73, 95% HDI for OR = [0.69, 0.77]) with increasing accuracy as grid coarseness decreased. As can be appreciated from *Figure 4*, there indeed seemed to be a cutoff value for contrast heterogeneity beyond which the figure could not be discriminated from the background. This cutoff may also explain why the interaction between session and grid coarseness was negligible. Below the cutoff point, the top and middle rows of *Figure 4* suggest that participants could discriminate the figure for increasingly coarser grids. Beyond the cutoff, however, grid coarseness seemed to have had no discernible effect regardless of how much training participants received. The characteristic triangular shape of the Arnold tongue thus gradually morphed into a rectangular shape.

Next, we examined simple effects of contrast heterogeneity on discrimination accuracy for each session separately (see *Table 1*). As expected from the presence of the cutoff, the effect of contrast heterogeneity increased over sessions, reflected in decreasing log-odds ($\beta$) and corresponding odds ratios (ORs) over sessions as shown in *Table 1*.

Finally, we evaluated whether the observed effects reflected localized learning in early visual cortex, as assumed by our model, implying that the training effect would be specific to the trained location. Performance in the transfer session should thus resemble that observed at training locations during early rather than late sessions. To test this, we estimated a hierarchical Bayesian logistic regression model with predictors for contrast heterogeneity, grid coarseness, session, their interactions, and an indicator for the transfer session. Subject-level random intercepts and slopes were included. From the fitted model, we generated posterior predictions of the population-level mean accuracy for each session. We then compared transfer (Session 9) with an early reference session in two complementary ways. First, we estimated the posterior probability that transfer session accuracy was lower than in the reference session. Second, we estimated the posterior probability that the difference between accuracy in the transfer and reference session lay within a region of practical equivalence (ROPE, ±2% accuracy). We used the second session, the earliest session after task familiarization, as reference. Our analysis revealed that performance in the transfer session was practically equivalent to session 2 (93% posterior probability of equivalence). Based on this, we expected that the local learning mechanism implemented in our model can provide quantitative predictions of performance changes over the course of the eight training sessions.

We evaluated the quantitative agreement between model synchrony and empirical discrimination performance. This analysis focused exclusively on synchrony. Rate-based readouts of our V1 model are not at all affected by variations in coupling strength (see also *Figure 4—figure supplement 1*). As such, they are insensitive to changes in coupling and are thus not viable as alternative mechanisms

to explain performance changes due to learning. To evaluate the quantitative agreement between model synchrony and empirical discrimination performance, we measured the similarity between simulated and behavioral Arnold tongues using Pearson correlations and weighted Jaccard similarity. Because we employed a leave-one-out cross-validation procedure, we obtained eight simulated Arnold tongues in sessions 2–8 after optimizing learning parameters on data from seven participants. Simulated Arnold tongues in each fold were always compared to behavioral Arnold tongues of the left-out participant. The first session did not involve learning, and model simulations were identical to those reported above. Note that data from the second session was used to adjust model parameters and hence only sessions 3–8 could be used for evaluating model predictions. This cross-validation approach enabled us to assess the model's ability to predict performance in unseen data, rather than merely fitting observed results post-hoc. *Figure 5a and b* show correlations and Jaccard similarity between simulated and behavioral Arnold tongues, respectively. Gray regions indicate a noise ceiling that was obtained by computing the fit between average behavioral Arnold tongues in a fold and the behavioral Arnold tongue of the left-out participant. The gray region marks the 25th to the 75th percentile of fit values obtained using this procedure. The figure demonstrates consistent quantitative agreement between simulated and behavioral Arnold tongues across sessions.

To examine this further, we tested whether the size of the simulated Arnold tongue across sessions was predictive of the size of the behavioral Arnold tongues. We quantified the size of each Arnold tongue in terms of the volume under its surface computed using Simpson's numerical integration. Arnold tongues grew across sessions with comparable growth curves for simulated and behavioral Arnold tongues (see *Figure 5c*). The precise relationship between simulated and behavioral Arnold tongue sizes is depicted in *Figure 5d*. Subsequently, we performed a Bayesian hierarchical linear regression to investigate this in sessions 3–8. We ignored the first two sessions since these were used for estimating model learning parameters. As expected, the size of the simulated Arnold tongue predicted the size of the behavioral Arnold tongue ($\beta$=0.54, 95% HDI [0.106, 0.935], Pr[$\beta$>0]=0.992). The model's capability to accurately reflect learning effects observed in human participants is consistent with the notion that enhanced synchrony among neural assemblies in early visual cortex resulting from perceptual learning enhances humans' ability to segregate figure from ground. This further strengthens the view that synchrony principles provide a viable neural grouping mechanism for texture segregation.

## Discussion

The role of synchrony in the gamma frequency band for visual perception remains a matter of debate (*Duecker et al., 2021*; *Fernandez-Ruiz et al., 2023*; *Ray and Maunsell, 2015*; *Roelfsema, 2023*). A putative role for gamma synchrony in processing the features of a stimulus both within and across visual areas (*Fries, 2009*; *Singer, 1999*; *Uhlhaas et al., 2008*; *Womelsdorf et al., 2007*) has been called into question based on the stimulus-dependence of gamma synchrony (*Ray and Maunsell, 2010*; *Ray and Maunsell, 2015*; *Roelfsema, 2023*). Alternatively, it has also been suggested that feature-dependent gamma frequencies and distance-dependent synchrony are key ingredients in a neural grouping mechanism underlying figure-ground segregation (*Lowet et al., 2015*; *Lowet et al., 2017*). It is well-established that the frequency of gamma oscillations in visual cortex depends on local stimulus features (*Baldi and Meir, 1990*; *Buia and Tiesinga, 2006*; *Hall et al., 2005*; *Henrie and Shapley, 2005*; *Roberts et al., 2013*; *Shapira et al., 2017*) and that lateral connectivity between neural groups within early visual cortex depends on cortical distance (*Amir et al., 1993*; *Boucsein et al., 2011*; *Eckhorn, 1994*; *Gilbert and Wiesel, 1989*; *Stettler et al., 2002*; *Ts'o et al., 1986*). It is likewise a well-known property of coupled oscillators that they synchronize when their coupling is sufficiently strong to overcome differences in their frequency, but not otherwise (*Acebrón et al., 2005*; *Ermentrout et al., 2019*; *Kuramoto, 1984*; *Neu, 1979*; *Strogatz, 2000*). Synchrony may thus drive the perceptual grouping of elements if they are sufficiently similar to each other within one image region, and thereby segregate it from other image regions based on their different levels of synchrony.

We tested this hypothesis in a psychophysics experiment wherein human observers discriminated the orientation of a texture-defined, rectangular figure region (vertical vs horizontal). The stimulus consisted of small Gabor annuli arranged on an irregular grid. Each Gabor annulus was characterized by its own contrast and the figure region was defined by less heterogeneous contrasts among the

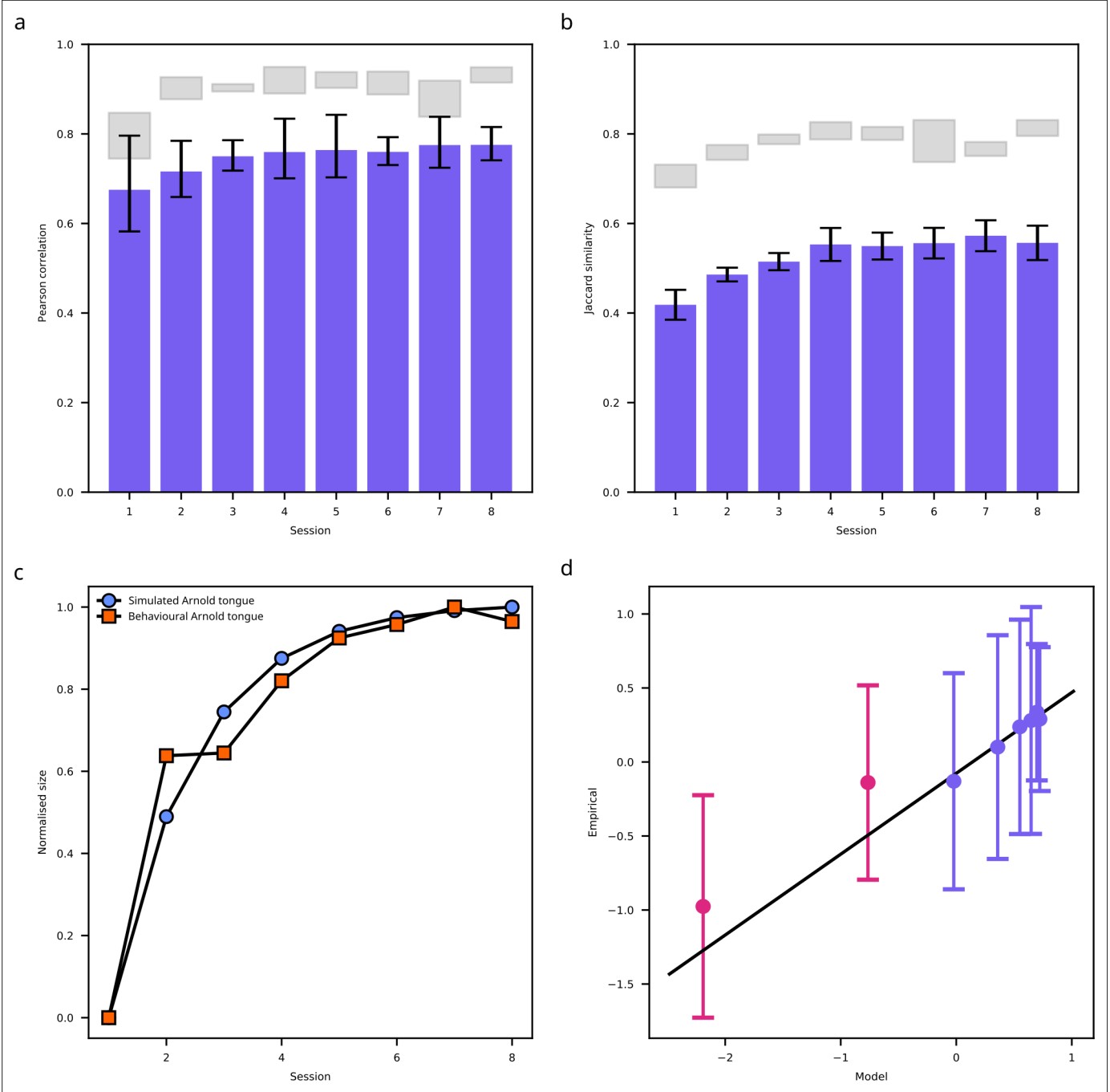

**Figure 5.** Model predictions of learning effects. (**a**) Pearson correlations between simulated and behavioral Arnold tongues for each training session (n = 8 participants). Error bars indicate 95% confidence intervals. Gray regions indicate a noise ceiling obtained by computing the fit between average behavioral Arnold tongues in a fold and the behavioral Arnold tongue of the left-out participant (25th to 75th percentile). (**b**) Weighted Jaccard similarity values between simulated and behavioral Arnold tongues for each training session (n = 8). Error bars and gray regions as in (**a**). (**c**) Sizes of simulated (blue circles) and behavioral (orange squares) Arnold tongues across sessions (n = 8). Arnold tongue sizes were averaged across participants and subsequently min-max normalized. This normalization highlights the growth patterns while accounting for the different value ranges of simulated and behavioral Arnold tongues. (**d**) Z-normalized sizes of behavioral Arnold tongues as a function of z-normalized sizes of simulated Arnold tongues (n = 8). Error bars indicate 95% confidence intervals. The black line represents the best-fitting regression from a Bayesian linear mixed-effects model with a random intercept for participants, fitted to data from training sessions 3–8 (blue circles).Red circles reflect data from the first two sessions that was not included in the mixed effect model. The black line was extended to include these points.

Gabor annuli compared to the background. We manipulated contrast heterogeneity and grid coarseness (distance between annuli) as a proxy of frequency detuning and coupling strength, respectively. Both contrast heterogeneity and grid coarseness affected discrimination accuracy. Specifically, we found that accuracies beyond 75% were limited to a triangular region in the space spanned by these two factors, forming a behavioral Arnold tongue. In line with our expectations, increased contrast heterogeneity in the figure permitted figure-ground segregation if accompanied by a reduction in grid coarseness. These results quantitatively aligned well with synchrony exhibited in a coupled oscillator V1 model exposed to the same texture stimuli.

The capacity of our model to predict human psychophysical performance is notable given that the key parameters of maximum coupling strength and coupling decay factor were obtained from neurophysiological data recorded from macaques (*Lowet et al., 2017*). This cross-species validation underscores the robustness of our model and suggests that the neural mechanisms underlying gamma oscillations and figure-ground segregation are largely conserved across primate species (*Buzsáki et al., 2013*). It is noteworthy that the parameter combination obtained from macaque data bordered the region of optimal combinations exhibiting the highest match to psychophysics results that our model could, in principle, achieve (see *Figure 3*). The slight deviation from the optimal regime likely stems from the fact that parameters were estimated from data obtained in macaques and subsequently used to predict human behavior. It is likely that horizontal connections in the human extend further than those in the macaque (*Amir et al., 1993*; *Burkhalter and Bernardo, 1989*; *Lund et al., 1993*; *Lund et al., 1993*; *Voges et al., 2010*) and may thus be associated with a slightly smaller coupling decay factor. Another possibility is that the parameter value we derived from *Lowet et al., 2017*, a study chosen because their paradigm targets the same TWCO components that guided our stimulus design, is an overestimate. As with any study, their data comes with uncertainty such that our estimates might not perfectly reflect actual decay rates. While we currently do not have alternative data to estimate the exact human decay factor and hence cannot establish how much model fit would be affected, any small to modest reduction would certainly further improve model fit.

To further investigate whether synchrony among neuronal populations exhibiting contrast-dependent frequencies provides a potential perceptual grouping mechanism, we tested whether training-induced changes of lateral coupling in a network of phase oscillators improved the readiness of these oscillators to synchronize and whether this model provided accurate predictions of performance on the figure-ground segregation task. We reasoned that for neural synchrony to function effectively as a grouping mechanism, it should be modifiable by experience in a manner that matches training-induced improvements in texture segregation. We observed that discrimination performance improved as a function of training session, in line with participants' growing experience with the stimuli. Importantly, we found that training-induced increases in accuracy were well accounted for by model predictions of synchrony strength inside the figure. Our results are consistent with the notion that synchrony mechanisms in low-level visual areas contribute in a behaviorally relevant manner to texture segregation and that training-induced changes of local synchrony are reflected by concurrent changes in perception. Synchrony and discrimination accuracy revealed highly congruent Arnold tongues. A close quantitative resemblance of these Arnold tongues was maintained across sessions as both tongues grew and changed form in a highly consistent manner. This supports the idea that learning-induced changes in figure-ground segregation may be mediated by plasticity-induced changes in synchrony. Oscillations have been shown to facilitate learning through spike-timing dependent plasticity (*Masquelier et al., 2009*), rendering an oscillation-based Hebbian learning mechanism biologically plausible.

It is important to note that the learning mechanism integrated into our model assumes that learning is local. We validated this assumption in the human participants by testing whether moving the figure region from its trained location to a new location would lead to transfer of performance to the new location, or rather a decrease in performance in the new location. Our results supported the latter. This is in line with other studies that demonstrated that after location-specific training, low-level visual areas contribute to the location and stimulus specificity of expert visual performance (*Karni and Bertini, 1997*). Based on previous findings that location-specific training induces localized plasticity in low-level visual areas (*Brosch et al., 2015*; *Raiguel et al., 2006*; *Schoups et al., 2001*; *Yang and Maunsell, 2004*), we further assumed that learning in our paradigm primarily affects lateral connectivity within V1 and hence manipulates coupling strength between neural assemblies. An alternative hypothesis

could be that learning, by targeting feedforward or feedback connectivity, alters the contrast sensitivity of neural assemblies. If this were to reduce the slope of the contrast-frequency relationship, it could theoretically offer a pathway to achieve synchrony across more heterogeneous contrasts by minimizing detuning rather than increasing coupling strength. However, empirical evidence suggests that training on perceptual tasks tends to steepen, rather than flatten, the contrast-frequency relationship (*Chen et al., 2013*; *Hua et al., 2010*; *Sanayei et al., 2018*). Given its lack of empirical support, we therefore did not incorporate this alternative mechanism into our model.

The predominant cue for figure-ground segregation in our stimuli lay in the global variations of population statistics in the contrast distribution, rather than local differences at the boundary between the figure and the ground (*De Weerd et al., 1994*; *Poort et al., 2016*; *Roelfsema et al., 2002*). This design was specifically chosen to preclude simple segregation based on mean firing rates. Nevertheless, our results indicate that a region comparison mechanism could still exploit firing rates to segregate figure from ground. While the firing rates of individual oscillators in our model are modulated by ongoing interactions, average firing rates in the figure region are insensitive to these interactions and hence purely driven by feedforward contrast extraction. By contrast, synchrony in our model arises from these interactions as they convert variance in local firing rates into coherence signals. Our results show that this can provide sufficient information for a subsequent read-out mechanism to distinguish figure from background. It might also provide additional information that downstream regions might exploit in addition to information carried by average firing rates. It might, for instance, provide a scaffold that can then be refined and read out by top-down mechanisms (*Ahissar and Hochstein, 1997*; *Ahissar and Hochstein, 2004*; *Hochstein and Ahissar, 2002*; *Liu and Weinshall, 2000*; *Rubin et al., 1997*). Such a scaffold might be compatible with widely accepted recurrent models in which boundary detection is followed by region-filling feedback (*Grossberg and Mingolla, 1985*; *Grossberg and Mingolla, 1987*; *Keil et al., 2005*; *Layton et al., 2014*; *Motoyoshi, 1999*; *Neumann et al., 2001*; *Pessoa and De Weerd, 2023*; *Roelfsema et al., 2002*), a notion substantiated by neurophysiological and psychophysical evidence (*Poort et al., 2016*; *Roelfsema et al., 2002*; *Self et al., 2012*) as well as by lesion and optogenetics experiments (*Kirchberger et al., 2021*; *Lamme et al., 1998*; *Supèr and Lamme, 2007*). We must note that we used the instantaneous frequencies of our model oscillators as a proxy for population firing rates. This is an oversimplification given that population firing rates are much lower than gamma (*Zachariou et al., 2021*). However, over the contrast range relevant to our stimuli, gamma frequency and population firing co-vary approximately linearly (*Zachariou et al., 2021*). Frequency thus served as a rate-like activation measure rather than a literal firing rate. It is, furthermore, important to acknowledge that our model does not account for attentional effects, although the significance of attention in figure-ground segregation and in learning is well-established (*Huang et al., 2020*) and it is likely that pure exposure to the stimuli in our experiment would have revealed very limited effects (*Seitz and Dinse, 2007*). Thus, while the current model indicates what early visual circuits could achieve in isolation, integrating the synchrony scaffold with rate-based mechanisms and attentional gain control remains a goal for future work.

A consideration to keep in mind in interpreting the effects of training on texture segregation is that participants at the outset of the experiment were unfamiliar with various aspects of the task unrelated to the perceptual challenge itself. They had to learn to maintain fixation, to establish stimulus-response mappings and associated decision processes, in addition to solving the perceptual challenge. As such, results in the first session may represent cognitive processes related to these non-perceptual factors. Future versions of our experiment might consider including a baseline training session during which participants get acquainted with the experimental setup and task using stimuli that define figure and background with features that are independent of those manipulated in the main experiment. Moreover, participants were not informed of which visual quadrant the figure would appear in the transfer session. This raises the concern that our results partly reflect visual search effects (*Eckstein, 2011*; *Neisser, 1964*) rather than a return to a naïve state of the figure-ground segregation skill. Arguably, however, this only affected a few trials and is thus insufficient to account for the loss of skill we observed. Furthermore, our model was designed to test the emergence of synchrony within the figure region itself, and as such, it did not include the background texture. While this approach allowed us to isolate the core mechanism of interest, it means our model provides an account of local grouping rather than a full simulation of figure-ground segregation. Finally, although a strength of this work is the prediction of human psychophysical performance based on a model whose parameters

were set by independent neurophysiological data, a weakness is the absence of neurophysiological data for our specific experimental paradigm. Such data would allow for a full mediation analysis from stimulus features via synchrony to behavior and could strengthen our interpretations. At the same time, a combined psychophysical and neurophysiological experiment in an animal model replicating the experimental conditions used here would benefit from strong predictions provided by the present study as well as prior neurophysiological data (*Lowet et al., 2017*). Despite these considerations and limitations, our results support the notion that gamma synchrony can serve a mechanistic role in figure-ground segregation.

The synchrony-based grouping mechanism studied here provides a theoretical framework for previous experimental results. A wide range of texture manipulations has been shown to drive segregation, including contrast (*Hadjipapas et al., 2015*), spatial frequency (*Bredfeldt and Ringach, 2002*; *Henriksson et al., 2008*), color (*Shapley and Hawken, 2011*), orientation (*Lamme, 1995*), and movement direction (*Lamme, 1995*). It is well documented that the difference between figure and background in one or a combination of these features (*Landy and Bergen, 1991*; *Motoyoshi and Nishida, 2001*; *Nothdurft, 1985a*; *Nothdurft, 1991b*; *Nothdurft, 1991a*) in population statistics (*De Weerd et al., 1992*; *Nothdurft, 1985b*) and in the physical proximity among texture elements within a figure (*De Weerd et al., 1992*; *Nothdurft, 1985b*) is the main parameters that determine the accuracy of figure-ground segregation. Much of this work consists of separate studies focusing on the contributions of single or restricted subsets of features to segregation. Viewed through the lens of TWCO, however, these features have their effect through the same mechanism. Most element features directly influence frequency detuning (*Dubey and Ray, 2020*; *Hadjipapas et al., 2015*; *Henrie and Shapley, 2005*; *Roberts et al., 2013*; *Shapira et al., 2017*), while proximity determines coupling strength via lateral connectivity in early visual cortex (*Boucsein et al., 2011*; *Gilbert and Wiesel, 1983*; *Lowet et al., 2015*; *Lowet et al., 2017*; *Stettler et al., 2002*; *Ts'o et al., 1986*). Rather than introducing a new explanatory variable, TWCO offers a mechanistic synthesis and shows how the established influence of these features on perception can emerge from the dynamics of coupled neural oscillators in V1. Thus, the success of these manipulations may arise precisely because they tap into the factors that determine whether synchrony can form among neural assemblies. As such, TWCO may provide a unifying principle that explains why these stimulus features are effective in modulating the efficiency of figure-ground segregation.

Future work should explore to what extent the principles of TWCO can explain segmentation of objects in natural images. While synchrony-based grouping mechanisms based on these principles have been used to segment natural images in machine vision (*Fang et al., 2014*; *Lowet et al., 2015*; *Nikonov et al., 2020*), it remains an open question whether cortical synchrony mediates human perception for such stimuli. Similarly, it remains an open question whether the principles outlined here generalize beyond the visual system to other sensory modalities. Interestingly, related forms of stimulus-dependent synchrony have been observed in auditory cortex, where it facilitates the integration of sound features and the segregation of auditory streams (*Giraud and Poeppel, 2012*). Finally, the principles of TWCO might provide a novel lens through which we can understand perceptual symptoms in neurological and psychiatric disorders. For example, schizophrenia is characterized by disrupted perceptual grouping and figure-ground segregation (*Liddle, 1987*; *Malaspina et al., 2004*; *Uhlhaas et al., 2006*) and disrupted visual gamma (*Spencer et al., 2003*). The prominent role of coupling strength within TWCO raises the possibility that reduced dendritic spine density in layer 3 of V1 within schizophrenia patients (*Fish et al., 2025*) may contribute to disrupted gamma synchrony and that this, in turn, may lead to disrupted perceptual grouping.

In conclusion, this study shows that figure-ground segregation performance can be well predicted by the factors that determine synchrony according to the theory of weakly coupled oscillators. Frequency detuning driven by contrast heterogeneity and coupling strength driven by physical distance may interact constructively to give rise to the perceptual skill of figure-ground segregation as well as its practice-induced enhancement. Our results show that a synchrony-based neural grouping mechanism can account for the observed behavioral patterns in a texture segregation task, and therefore remains a viable explanation for figure-ground segregation that cannot be ruled out. The documented dependence of gamma synchrony on stimulus features and element distance is essential components rather than obstacles to such a mechanism. This research sheds additional light on the underlying mechanisms of visual perception and perceptual learning and suggests that

gamma oscillations and synchrony may be involved in the training-induced enhancement of figure-ground segregation.

# Methods
## Behavioral experiments
The study and its experimental procedures were approved by the local Ethical Committee of the Faculty of Psychology and Neuroscience (ERCPN; ERCPN-176_02_07_2006_V2_A1).

### Participants
Eight healthy volunteers (six female, mean age = 23.75, standard deviation = 6.4536) participated in this study. Our study employed a repeated-measures design with extensive sampling, collecting a large number of trials from each participant. Sample size was determined based on comparable studies investigating visual perception and perceptual learning in humans (*Intoy et al., 2024*; *Lange et al., 2020*; *Tesileanu et al., 2020*). All participants had normal or corrected-to-normal visual acuity. After receiving full information about all procedures and the right to withdraw participation at any time, participants gave their written informed consent. All participants were compensated monetarily for their time.

### Stimuli
Each texture stimulus consisted of a full-screen irregular grid of non-overlapping Gabor annuli with a diameter 0.7°, a spatial frequency of 5.7 cycles/degree and a mean luminance of $60.76\,\mathrm{Cd/m^2}$ placed on a gray ($60.76\,\mathrm{Cd/m^2}$) background. Annuli contrasts were uniformly sampled from the full contrast range $U\left[0,1\right]$, except for a rectangular figure region [(9±0.7)°× (5±0.4)°] whose contrasts were drawn from a second uniform distribution $U\left[0.5 - \frac{\zeta}{2}, 0.5 + \frac{\zeta}{2}\right]$ with range $\zeta$ whose values were{0.01,0.2575, 0.505,0.7525,1}. The figure region thus exhibited limited contrast heterogeneity, except when $\zeta = 1$ which is identical to the background (maximum) contrast heterogeneity. The center of the figure region was placed at an eccentricity of (7±1)°. The polar angle of the figure was varied on each trial with the condition that it was always completely inside a single visual field quadrant. The coarseness of the grid was expressed as a factor $\rho$ that scales the average center-to-center distance between any pair of neighboring annuli in the whole texture. The values of $\rho$ were {1,1.125, 1.250, 1.375, 1.5} Each annulus was initially placed on a regular grid and subsequently slightly shifted in a random direction by a distance chosen from a uniform distribution that ranged from zero to half of the edge-to-edge distances of neighboring annuli. All combinations of $\zeta$ and $\rho$ yield 25 unique stimulus conditions.

### Tasks and procedure
The experiment consisted of nine consecutive sessions (eight training and one transfer session) with a two-alternative forced choice design in which participants were required to indicate whether the rectangular figure was oriented horizontally or vertically by pressing the right and left arrow key, respectively. Responses were given with the middle and index fingers of the right hand. Each trial of the experiment started with the presentation of a fixation point (a small bright turquoise disk of 2°×2°) for minimally 1000 ms, during which accurate fixation was to be initiated (i.e. deviation <2° from fixation point) to trigger stimulus presentation. Participants were required to maintain fixation throughout presentation of the stimulus (1000 ms or less in case that a participant lost fixation or provided a response). Participants received feedback after each trial in the form of color changes (green correct; red incorrect) of the fixation point lasting for 500 ms. Feedback was followed by a 600 ms inter-trial interval during which an isoluminant (gray) screen was shown. When a participant's gaze fell outside the fixation window during the fixation period preceding the stimulus, or during stimulus presentation, the trial was aborted. Aborted trials were repeated at a randomly chosen time during the experiment.

   The 25 conditions defined by contrast heterogeneity and grid coarseness were aggregated into experimental blocks such that all 25 combinations were shown exactly once per 25-trial block in random order. Each participant completed 30 blocks (750 trials) in each of the sessions. The figure was placed in the lower right quadrant for the eight training sessions. In the transfer session, the figure was

moved to the orthogonal (upper left) quadrant. Participants were made aware of the figure displacement but were not told in which quadrant to expect it.

The experiment was conducted in a dimly lit room. A chin and headrest were used to support the participant's head and to keep eye-screen distance constant at 57 cm. Stimuli were displayed on a 19 Samsung SyncMaster 940BF LCD monitor (Samsung, Seoul, South Korea; 60 Hz refresh rate, 1280 × 1024 resolution). Stimulus representation and response recording were performed by Psychtoolbox-3 for MATLAB 64-Bit (Version 3.0.14 - Build date: April 6, 2018), under Microsoft Windows. Fixation was monitored with a desktop-mounted Eyelink 1000 eye-tracker (SR Research Ltd., 500 Hz or 1000 Hz sampling frequency, 0.01° RMS spatial resolution, eye-movement data were down-sampled to 250 Hz).

## Statistical analyses

We used Bayesian hierarchical regression to analyze main and interactions of variables of interest which include manipulated stimulus features (contrast heterogeneity, grid coarseness, and their interaction), model synchrony, and learning effects (session and its interactions with stimulus features). Stimulus features and model synchrony were z-scored and session was mean-centered. Each of these statistical models included subject-specific intercepts and slopes for all predictors. To investigate the relationship between the sizes of empirical and model Arnold tongues, we used Bayesian hierarchical regression with subject-level random intercepts. Because tongue size data entail only one measurement per subject per session, there was insufficient information to estimate subject-level slopes reliably.

All analyses were conducted using Bambi (v0.15.0) and ArviZ (v0.22.0) in Python. Priors were weakly informative defaults. Specifically, fixed effects were given Normal(0, 2.5), intercepts Normal(0, 2.5), and group-level standard deviations HalfNormal(2.5) priors. Correlations among random slopes and intercepts were given an LKJ(1) prior. Models were estimated using the No-U-Turn Sampler (NUTS) with 4 chains of 2,000 draws each, following a 2000-draw tuning phase, for a total of 8000 posterior samples. For all Bayesian models, convergence was assessed using standard diagnostics. All R values were approximately 1.00, and all effective sample sizes (ESS) were sufficient, with the smallest ESS across analyses being 2200.

## Oscillator model of V1

We model a small patch of V1 that receives input from a 6.7° ×6.7° square region of the visual field. The area of this square region matches the area of the rectangular figure region in our psychophysics experiments. The center of this region is furthermore located at an eccentricity matching that of the figure. We model this V1 patch as a network of weakly coupled phase oscillators arranged on an $n \times n (n = 20)$ irregular grid on the cortical surface. To that end, we first defined a regular grid of receptive field centers for each oscillator in visual space and subsequently transformed receptive field coordinates to cortical coordinates of V1 using a complex-l (***Balasubramanian and Schwartz, 2002***; ***Schwartz, 1980***) with generic human parameter values ($a = 0.7, \alpha = 0.9$; ***Polimeni et al., 2005***). While receptive fields are thus equally spaced in the visual field, neural oscillators themselves are not equally spaced on the cortical surface. The phase of each neural oscillator evolves according to a Kuramoto model:

$$\dot{\theta}_i = \omega_i + \frac{1}{N} \sum_{j=1}^{N} K_{ij}^s sin\left(\theta_j - \theta_i\right), i = 1, \ldots, N; s = 1, \ldots, 9 \tag{1}$$

where, $\theta_i$ is the phase of the $i$ th oscillator, $\omega_i$ its intrinsic frequency, $K_{ij}^s$ the coupling strength between oscillators $i$ and $j$ in session $s$ (note that $s$ is an index and not an exponent) and $N = n^2$ is the total number of oscillators. We treat the instantaneous frequency $\frac{\dot{\theta}}{2\pi}$ of each oscillator as a proxy for the instantaneous population firing rate of the corresponding neural assembly.

## Intrinsic frequency

In accordance with electrophysiological findings, the intrinsic frequency of each oscillator is a function of the local contrast in its receptive field (***Roberts et al., 2013***). Specifically, the typical oscillation frequency $\nu$ (in $Hz$ with corresponding $\omega = 2\pi\nu$) of a neural circuit in V1 is a linear function of local contrast (***Lowet et al., 2015***):

$$\nu = 25 + 0.25C \tag{2}$$

The local contrast received by each oscillator $i$ is given by the weighted root-mean-squared (RMS) value of contrast (*Frazor and Geisler, 2006*):

$$C_i = \sqrt{\sum_{h=1}^{M} w_{ih} \frac{\left(L_h - \bar{L}\right)^2}{\bar{L}^2} / \sum_{h=1}^{M} w_{ih}} \tag{3}$$

where $L_h$ is the luminance of pixel $h$ in the stimulus, $\bar{L}$ is the mean luminance over all pixels, and $w_{ih}$ is the weight of pixel $h$ and oscillator $i$. The weighting was specific to each oscillator as it reflects its unique receptive field which we modeled using an isotropic 2D Gaussian function:

$$w_{ih} = exp\left(-\frac{\left(x_h - X_i\right)^2 + \left(y_h - Y_i\right)^2}{2\sigma_i^2}\right), \tag{4}$$

Here, $(x_h, y_h)$ are the coordinates of the $h^{\text{th}}$ pixel, while $(X_i, Y_i)$ are the coordinates of the receptive field center of the $i$ th oscillator. In addition, $\sigma_i$ is the size of the receptive field. We estimated receptive field sizes based on their location relative to the center of gaze. Specifically, receptive field diameter in V1 exhibits a threshold linear relationship with receptive field eccentricity ($e$; *Freeman and Simoncelli, 2011*) such that $\emptyset = max\left(0.172e - 0.25, 1\right)$. We related the receptive field diameter to the standard deviation of a Gaussian in two steps. First, we related the diameter to the full width at half maximum (FWHM) of a Gaussian beam $FWHM = \frac{\sqrt{ln2}}{\sqrt{2}}\emptyset$ (*Hill, 2007*). Then, we related the FWHM to the standard deviation $\sigma = \frac{FWHM}{2\sqrt{2ln2}}$. Combining these steps, the standard deviation is one fourth of the receptive field diameter.

## Adaptive coupling

The coupling strength $K_{ij}^1$ between pairs of oscillators in the first session is a function of their cortical distance:

$$K_{ij}^1 = \gamma e^{-\lambda d_{ij}}. \tag{5}$$

Here, $\gamma$ is the maximum coupling strength and $\lambda$ controls how fast the coupling strength decreases as a function of cortical distance $d_{ij}$ between oscillators $i$ and $j$. We estimated $\gamma$ and $\lambda$ from previously published data relating coupling strength to cortical distance within V1 in two macaque monkeys (*Lowet et al., 2017*).

Coupling strength in the remaining sessions is the result of an offline learning process that takes the experience accumulated over an individual training session into account. Specifically, learning in our model depends on the pairwise phase-locking value (PLV; *Lachaux et al., 1999*) between model oscillators accumulated over trials within one session. Phase-locking values were computed over the second half of the simulation period, which was subsampled to 50 timepoints. Accumulation across trials involves summing PLVs over trials, where the contribution of each trial is weighted by the probability that the model would produce a correct response on that trial. The weighted PLV is summarized in a matrix $Q$. To obtain the probability of a correct response ($P_c$) from model simulations, we related it to the degree of synchrony ($r$) among phase oscillators through a psychometric function:

$$P_c = 1/\left[1 + \exp\left(-\mu_0 - \mu_1 r\right)\right] \tag{6}$$

Parameters of this function (i.e. $\mu_0$ and $\mu_1$) were estimated based on model simulations and empirical results from the first session. The temporal evolution of pairwise coupling strength is given by a Hebbian-type learning rule:

$$\dot{K}_{ij} = \epsilon\left(\gamma Q_{ij}^s - K_{ij}\right). \tag{7}$$

Here, $\epsilon$ is a learning rate. Essentially, pairwise structural coupling approaches pairwise functional coupling, as measured by the weighted PLV within a session ($Q^s$), scaled by the maximum coupling strength $\gamma$. Integration of *Equation 5* with respect to time yields

$$K_{ij}^{s+1} = exp\left(-\epsilon t\right) K_{ij}^{s} + \left[1 - exp\left(-\epsilon t\right)\right] \gamma Q_{ij}^{s}. \tag{8}$$

Here, $t$ is the time between two sessions during which learning occurs (e.g., during sleep). Since neither $t$ nor $\epsilon$ can be measured independently and are not known a priori, we merged them into a single free parameter $E = \epsilon t$. We refer to this as the effective learning rate. We adjusted the parameter $E$ to maximize the correspondence, measured by the weighted Jaccard similarity, between the distribution of performance observed in the *second* experimental session and the distribution of synchrony after letting the model learn according to *Equation 6*. To that end, we used a coarse-to-fine grid search wherein we let the model learn using a grid of 25 candidate effective learning rates and selected the value that enabled best prediction of session 2 performance. We then created a new, finer grid around the best effective learning rate and repeated this procedure. In total, we explored five nested grids. Note that the learning procedure depends on data from the first two sessions to establish a mapping from synchrony to performance (parameters $\mu_0$ and $\mu_1$ of the psychometric function linking synchrony to performance) and to estimate the effective learning rate, respectively. We kept these parameters fixed for predicting the results of sessions 3–8. To further disentangle data used for parameter tuning and data used for testing model predictions, we utilized a leave-one-out cross-validation procedure. We estimated all parameters from the first two sessions of seven of our eight participants and then predicted results of session 3–8 in the left-out participant. We repeated this procedure eight times, once per participant, and stored all results for further analysis.

## Simulations

We simulated eight training sessions, each consisting of 30 blocks with 25 trials. Within each trial, we simulated a one-second stimulus monitoring interval assigned to a specific combination of contrast heterogeneity and grid coarseness. All simulations were performed in Python 3.12.2 using the odeint method from scipy's (version 1.12.0) integrate submodule. For each simulated trial, we evaluated synchrony by measuring the radius ($r \in \left[0, 1\right]$) of the Kuramoto order parameter given by

$$re^{i\psi} = \frac{1}{N}\sum_{j=1}^{N} e^{i\theta_j} \tag{9}$$

where $\theta_j$ is the phase of the oscillator $j$. For each simulated trial, $r$ was averaged within the second half of the trial duration, and over all blocks.

## System specifications

All analyses and simulations were performed as a Docker containerized Snakemake workflow executed on a single compute node of Maastricht University's Data Science Research Infrastructure (DSRI). The node is equipped with two AMD EPYC 7551 32-Core Processors, has a nominal 512 GB of RAM, and operates on Fedora 37. The workflow utilized 30 of 64 available cores to simulate all blocks of a particular trial in parallel. To ensure that all results can be reproduced exactly, the random seed of our workflow was fixed at 1709026616.

## Code availability

The code for data acquisition can be accessed at https://github.com/ccnmaastricht/TextureStimuli-FigureGround, copy archived at *ccnmaastricht, 2024*. The code for performing all analyses and simulations can be accessed at https://github.com/ccnmaastricht/NeuralSynchrony-FigureGround, copy archived at *ccnmaastricht, 2026*.

## Additional information

### Funding
No external funding was received for this work.

## Author contributions
Maryam Karimian, Conceptualization, Data curation, Investigation, Visualization, Methodology, Writing – original draft, Writing – review and editing; Mark Jonathan Roberts, Data curation, Writing – original draft, Writing – review and editing; Peter De Weerd, Conceptualization, Supervision, Writing – original draft, Project administration, Writing – review and editing; Mario Senden, Conceptualization, Formal analysis, Supervision, Investigation, Visualization, Methodology, Writing – original draft, Project administration, Writing – review and editing

## Author ORCIDs
Maryam Karimian ⓘ https://orcid.org/0000-0001-7436-0787
Mark Jonathan Roberts ⓘ https://orcid.org/0000-0001-7513-1281
Peter De Weerd ⓘ https://orcid.org/0000-0003-2252-5548
Mario Senden ⓘ https://orcid.org/0000-0002-5598-6167

## Ethics
All participants received full information about all procedures and the right to withdraw participation at any time. Participants gave their written informed consent and consent to publish. All participants were compensated monetarily for their time. The study and its experimental procedures were approved by the local Ethical Committee of the Faculty of Psychology and Neuroscience (ERCPN) under identifier ERCPN-176_02_07_2006_V2_A1.

Reviewer #1 (Public review): https://doi.org/10.7554/eLife.105482.3.sa1
Reviewer #2 (Public review): https://doi.org/10.7554/eLife.105482.3.sa2
Author response https://doi.org/10.7554/eLife.105482.3.sa3

---

# Additional files

## Supplementary files
Supplementary file 1. Design analysis of main analysis in session 1. Detection probability refers to the proportion of simulated datasets in which the posterior probability of an effect exceeded 0.95 in the predicted direction. Type-S error indicates the probability of detecting an effect, but in the wrong direction (sign reversed). Type-M error refers to the ratio of estimated to true effect size when detected. A value of 1 indicates no deviation, whereas values larger (smaller) than 1 indicate that effects are over (under) estimated.

MDAR checklist

## Data availability
All data generated or analyzed during this study are openly accessible at https://doi.org/10.5281/zenodo.10817187.

The following dataset was generated:

| Author(s) | Year | Dataset title | Dataset URL | Database and Identifier |
|---|---|---|---|---|
| Karimian M, Roberts MJ, De Weerd P, Senden M | 2024 | Human Psychophysics Dataset on Figure Ground Segregation in Texture Stimuli | https://doi.org/10.5281/zenodo.10817187 | Zenodo, 10.5281/zenodo.10817187 |

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
