## [Editor Report · eLife Assessment]

Karimian et al. present a **valuable** new model to explain how gamma-band synchrony (30-80 Hz) can support human visual feature binding by selectively grouping image elements, countering recent criticisms that the stimulus dependence of gamma oscillations limits their functional role. Grounded in the theory of weakly coupled oscillators the model captures behavioural patterns observed in human psychophysics, offering support for the potential role of synchrony-based mechanisms in feature-binding. The development of the model in alignment with primate electrophysiology **convincingly** supports the paper's claims that gamma synchrony may be the underlying mechanism. While the paper does not present electrophysiological results that directly link gamma oscillations to figure-ground segregation in the presented task, the model makes several predictions that can be tested experimentally.

---

## [Referee Report · Reviewer #1 (Public review)]

Summary:

This paper by Karimian et al proposes an oscillator model tuned implementing binding by (gamma) synchrony principles in a visual task. The authors set out to show how well these principles explain human behavior in a figure-ground segregation tasks. The model is inspired by electrophysiological findings in non-human primates suggesting that gamma oscillations in early visual cortex implement feature-binding through a synchronization of feature-selective neurons. The psychophysics experiment involves the identification of a figure consisting of gabor annuli, presented on a background of gabor annuli. The participants' task is to identify the orientation of the figure. The task difficulty is varied based on the contrast and density of the gabor annuli that make up the figure. The same figures are used as inputs to the oscillator model. The authors report that both the discrimination accuracy in the psychophysics experiment and the synchrony of the oscillators in the proposed model follow a similar "Arnold Tongue" relationship when depicted as a function of the texture-defining features of the figure. This finding is interpreted as evidence for gamma synchrony being the underlying mechanism of the figure-ground segregation.

Strengths:

The design of the proposed model is well-informed by electrophysiological findings, and the idea of using computational modeling to bridge between intracranial recordings in non-human primates and behavioral results in human participants is interesting. Previous work has criticized the gamma synchrony theories based on the observation that synchronization in the gamma-band is highly localized and the frequency of the oscillation depends on the visual features of the stimulus. I appreciate how the authors demonstrate that frequency-dependence and local synchronization can be features of gamma synchrony, and not contradictory to the theory. As such, I feel that this work has the potential to contribute meaningfully to the debate on whether binding by gamma synchrony is a biophysically realistic model of feature-binding in visual cortex.

I also acknowledge the additional simulations the authors present in this version of the manuscript, showing that the model is able to segregate figure from ground.

Weaknesses:

The authors have addressed my previous concerns regarding the quantification of effect sizes. I also appreciate the authors argument that the results support the idea of feature-binding through synchronization in the gamma-band, as the model's parameters were informed by electrophysiological recordings from non-human primates. Personally, I would have been curious to see if the intrinsic frequencies of the model are indeed in the gamma-band, I don't believe the authors include a figure on that. Weaknesses are still the absence of electrophysiological recordings to support the frequency-specificity of the claims, e.g. in the form of EEG/MEG recordings, but I understand that these may be difficult to obtain, as gamma oscillations are relatively weak in response to static gratings. As the authors emphasize in this updated version, they present one possible mechanism of feature binding that is not contrasted to alternative mechanisms such as binding by increased firing rates. Understandably, implementing a second model would be out of scope.

The presented simulations and behavioural results support the authors aim of presenting an oscillator model informed by gamma synchronization in V1 that supports figure-ground segregation.

Likely impact:

This work makes several predictions about the degree of synchronization for different visual properties of the figure, that could be tested with electrophysiological methods. I therefore believe that the paper has the potential to motivate interesting follow-up studies to understand how visual cortex solves the binding problem.

Comment on revised version:

In this reviewed version of the manuscript, the authors present several follow-up simulations and clarifications that address previously outlined weaknesses.

---

## [Referee Report · Reviewer #2 (Public review)]

The authors aimed to investigate whether gamma synchrony serves a functional role in figure-ground perception. They specifically sought to test whether the stimulus-dependence of gamma synchrony, often considered a limitation, actually facilitates perceptual grouping. Using the theory of weakly coupled oscillators (TWCO), they developed a framework wherein synchronization depends on both frequency detuning (related to contrast heterogeneity) and coupling strength (related to proximity between visual elements). Through psychophysical experiments with texture discrimination tasks and computational modeling, they tested whether human performance follows patterns predicted by TWCO and whether perceptual learning enhances synchrony-based grouping.

Strengths:

(1) The theoretical framework connecting TWCO to visual perception is innovative and well-articulated, providing a potential mechanistic explanation for how gamma synchrony might contribute to both feature binding and separation.

(2) The methodology combines psychophysical measurements with computational modeling, with a solid quantitative agreement between model predictions and human performance.

(3) In particular, the demonstration that coupling strengths can be modified through experience is remarkable and suggests gamma synchrony could be an adaptable mechanism that improves with visual learning.

(4) The cross-validation approach, wherein model parameters derived from macaque neurophysiology successfully predict human performance, strengthens the biological plausibility of the framework.

Likely Impact and Utility:

This work offers a fresh perspective on the functional role of gamma oscillations in visual perception. The integration of TWCO with perceptual learning provides a novel theoretical framework that could influence future research on neural synchrony.

The computational model, with parameters derived from neurophysiological data, offers a useful tool for predicting perceptual performance based on synchronization principles. This approach might be extended to study other perceptual phenomena and could inspire designs for artificial vision systems.

The learning component of the study may have a particular impact, as it suggests a mechanism by which perceptual expertise develops through modified coupling between neural assemblies. This could influence thinking about perceptual learning more broadly, but also raises questions about the underlying mechanism.

Additional Context:

Historically, the functional significance of gamma oscillations has been debated, with early theories of temporal binding giving way to skepticism based on gamma's stimulus-dependence. This study reframes this debate by suggesting that stimulus-dependence is exactly what makes gamma useful for perceptual grouping.

The successful combination of computational neuroscience and psychophysics is a significant strength of this study.

The field would benefit from future work extending (if possible) these findings to more naturalistic stimuli and directly measuring neural activity during perceptual tasks. Additionally, studies comparing predictions from synchrony-based models against alternative mechanisms would help establish the specificity of the proposed framework.

Comments on revised version:

The authors now soften their claim. However, the paper demonstrates that TWCO-derived predictions quantitatively match human figure-ground perception in texture stimuli, and that a synchrony-based readout provides a viable mapping from stimulus to behavior. Given that they cite (and do not show in this paper) the link to synchrony, what they actually establish is that this particular transformation of stimulus features maps better onto behavior. That's meaningful, but it is not a demonstration of mechanism.

---

## [Author Response]

The following is the authors’ response to the original reviews.

**Public Reviews:**

**Reviewer #1 (Public review):**
Summary:This paper by Karimian et al proposes an oscillator model tuned to implement binding by synchrony (BBS*) principles in a visual task. The authors set out to show how well these BBS principles explain human behavior in figure-ground segregation tasks. The model is inspired by electrophysiological findings in non-human primates, suggesting that gamma oscillations in early visual cortex implement feature-binding through a synchronization of feature-selective neurons. The psychophysics experiment involves the identification of a figure consisting of gabor annuli, presented on a background of gabor annuli. The participants' task is to identify the orientation of the figure. The task difficulty is varied based on the contrast and density of the gabor annuli that make up the figure. The same figures (without the background) are used as inputs to the oscillator model. The authors report that both the discrimination accuracy in the psychophysics experiment and the synchrony of the oscillators in the proposed model follow a similar "Arnold Tongue" relationship when depicted as a function of the texture-defining features of the figure. This finding is interpreted as evidence for BBS/gamma synchrony being the underlying mechanism of the figure-ground segregation.Note that I chose to use "BBS" over gamma synchrony (used by the authors) in this review, as I am not convinced that the authors show evidence for synchronization in the gamma-band.

We thank the reviewer for their careful assessment of our manuscript and useful comments that we believe have served to strengthen our work.

Strengths:The design of the proposed model is well-informed by electrophysiological findings, and the idea of using computational modeling to bridge between intracranial recordings in non-human primates and behavioral results in human participants is interesting. Previous work has criticized the BBS synchrony theory based on the observation that synchronization in the gamma-band is highly localized and the frequency of the oscillation depends on the visual features of the stimulus. I appreciate how the authors demonstrate that frequency-dependence and local synchronization can be features of BBS, and not contradictory to the theory. As such, I feel that this work has the potential to contribute meaningfully to the debate on whether BBS is a biophysically realistic model of feature-binding in visual cortex.Weaknesses:I have several concerns regarding the presented claims, assessment of meaning and size of the presented effects, particularly with regard to the absence of a priori defined effect sizes.Firstly, the paper makes strong claims about the frequency-specificity (i.e., gamma synchrony) and anatomical correlates (early visual cortex) of the observed effects. These claims are informed by previous electrophysiological work in non-human primates but are not directly supported by the paper itself. For instance, the title contains the word "gamma synchrony", but the authors do not demonstrate any EEG/MEG or intracranial data in from their human subjects supporting such claims, nor do they demonstrate that the frequencies in the oscillator model are within the gamma band. I think that the paper should more clearly distinguish between statements that are directly supported by the paper (such as: "an oscillator model based on BBS principles accounts for variance in human behavior") and abstract inferences based on the literature (such as "these effects could be attributed to gamma oscillations in early visual cortex, as the model was designed based on those principles").

We thank the reviewer for this helpful comment and agree that the scope of our claims should be clearly delineated between what is directly supported by our data and what is theoretically inferred from prior literature.

We revised the Abstract, Introduction, and early Discussion to moderate the strength of our statements and make the distinction explicit. The revised title now emphasizes that our study tests principles derived from prior work on gamma synchrony rather than directly demonstrating gamma activity in humans. Throughout the text, we use more cautious phrasing that highlights potential mechanisms and theoretical predictions. The intention of our study was not to position synchrony as the only viable mechanism of figure–ground perception. Rather, our goal was to reinvigorate it as a potential contender by showing that features often cited as limitations of synchrony-based binding may in fact be essential properties of the mechanism. We updated phrasing throughout the manuscript to make this clearer and avoid overstating the study’s contribution.

Importantly, our model is not agnostic with respect to frequency band. Oscillator frequencies exhibited by model units are within the gamma range by design. Frequency emerges directly from the contrast within each oscillator’s receptive field, following an empirically established relationship between stimulus contrast and gamma frequency. To our knowledge, such a robust, quantitative relationship between stimulus features to exact oscillation frequency has not been consistently demonstrated for other frequency bands. This relationship yields gamma-band frequencies for all contrasts used in our simulations. The model is thus indeed a gamma oscillator model of V1, not a generic instantiation of Binding by Synchrony (BBS) principles.

That said, we fully agree with the reviewer that our study cannot demonstrate a direct link between gamma synchrony in visual cortex and human behavior. Our behavioral and modeling results instead show that synchronization principles derived from gamma-band physiology in V1 can predict perceptual performance patterns. We now make this distinction explicit throughout the revised manuscript.

Secondly, unlike the human participants, the model strictly does not perform figure-ground segregation, as it only receives the figure as an input.

We thank the reviewer for the opportunity to clarify our modeling approach. We chose not to model the background to reduce computational cost, since including it requires a substantially larger number of oscillators without changing the model’s predictions. The model thus indeed only receives the figure region as input. We aimed to test the local grouping mechanism predicted by TWCO, rather than to simulate a full figure–ground segregation process including a read-out stage. Our model therefore isolates the conditions under which local synchrony emerges within the figure region, assuming that a downstream read-out mechanism (not explicitly modeled here) would detect regions of coherent activity. The exact nature of such a read-out mechanism was beyond the scope of our work.

To confirm that our simplified model is a valid proxy, we ran additional simulations including the background and found that a coherent figure assembly reliably emerges, as can be seen in the phase-locking patterns relative to a reference oscillator at the center of the figure. This validates that the principles of local grouping we studied in isolation hold even when the figure is embedded in a noisy surround. We have added an explicit note in the Results (paragraph 2) that we only simulate the figure and added Supplementary Figure S1 showing the additional simulations.

Finally, it is unclear what effect sizes the authors would have expected a priori, making it difficult to assess whether their oscillator model represents the data well or poorly. I consider this a major concern, as the relationship between the synchrony of the oscillatory model and the performance of the human participants is confounded by the visual features of the figure. Specifically, the authors use the BBS literature to motivate the hypothesis that perception of the texture-defined figure is related to the density and contrast heterogeneity of the texture elements (gabor annuli) of the figure. This hypothesis has to be true regardless of synchrony, as the figure will be easier to spot if it consists of a higher number of high-contrast gabors than the background. As the frequency and phase of the oscillators and coupling strength between oscillators in the grid change as a function of these visual features, I wonder how much of the correlation between model synchrony and human performance is mediated by the features of the figure. To interpret to what extent the similarity between model and human behavior relies on the oscillatory nature of the model, the authors should find a way to estimate an empirical threshold that accounts for these confounding effects. Alternatively, it would be interesting to understand whether a model based on competing theories (e.g., Binding by Enhanced Firing, Roelfsema, 2023) would perform better or worse at explaining the data.

We thank the reviewer for these insightful and constructive comments, which have prompted additional analyses that we believe substantially strengthen our work. The reviewer raises two main points: (1) the need for a benchmark to assess our model’s performance, and (2) the concern that the relationship between model synchrony and behavior might be a non-causal “confound” of the visual features. We address each point below.

(1) Benchmarking model performance

We agree that it is important to assess how well our model performs relative to the data and included this in the original manuscript. We did not predefine an absolute good fit threshold because absolute agreement depends on irreducible noise and inter-subject variability, making a universal cutoff arbitrary. Instead, we had benchmarked model performance in two complementary ways. First, the noise ceiling shown in Figure 5 provides an empirical benchmark for the maximum fit any model could achieve on our data. Simulated Arnold tongues (based on synchrony) approach this ceiling achieving 89% of possible similarity for correlation and 79% of possible similarity for weighted Jaccard similarity, respectively. Second, the parameter sweep (Figure 3) situates our model’s performance within the broader parameter space. It shows that the model, whose key parameters were fixed a priori from independent macaque neurophysiological data, lies close to the optimal regime for explaining the human data. It also provides an estimate of the lower bound (worst-performing point) on the fit that a misspecified model implementing the identical mechanism would achieve. Our model with fixed a priori parameters does 1.41 times better than a misspecified model for the correlation fit metric and 3 times better for weighted Jaccard similarity.

(2) Synchrony as mechanism vs. potential confound

We appreciate the reviewer’s suggestion to test whether synchrony explains behavior beyond stimulus features. In our framework, synchrony is a near-deterministic function of the manipulated stimulus features given fixed model parameters. As a result, synchrony and the stimulus features are collinear (R^2^≈0.8) leaving no independent variance for synchrony to explain once stimulus features are included. Adding both into one statistical model yields unstable coefficients and no out-of-sample improvement.

Mechanistically, we believe the relevant question is not whether synchrony explains behavior beyond stimulus features but whether synchrony is the correct transformation of the stimulus features to reproduce the behavioral pattern. Please note that in our design we ensured that mean contrast and luminance are identical in the figure and the background such that there are not more high-contrast Gabors in the figure than in the background. We did this with the aim to render mean contrast not a relevant feature. However, there are more high-contrast Gabors in the background, and it is conceivable that the absence of such high contrasts in the figure drives the detection/discrimination of the figure. We therefore agree that testing alternative models would further clarify the unique explanatory value of the synchrony mechanism. To that end, we derived two alternative rate-based readouts from the same V1 simulations of our model from which we derived synchrony. First, average firing rates inside the figure and second, the difference between average firing rates inside the figure and average firing rates in the background (rate difference). We analyzed each individually as predictors of behavior and performed a model comparison based on out-of-sample predictions. While rate difference (but not average firing) showed meaningful associations with performance when considered alone, the synchrony readout had a larger effect size and was favored by the model comparison. We added a new subsection comparing synchrony to rate-based alternatives in the Results (paragraphs 7-9), including additional Bayesian analyses and LOO-CV model comparison. Please note that the model comparison we added to the manuscript provides an additional benchmark beyond the map-level ceiling analysis. It indicates that the mapping from stimulus features to behavior via synchrony generalizes best without requiring an a priori good-fit threshold.

We agree that formally comparing our model to a sophisticated rate-based alternative, such as an instantiation of the Binding by Enhanced Firing model, is an important direction for future work. However, it remains an open and non-trivial question whether such a model could quantitatively reproduce the precise shape of the behavioral Arnold tongue that emerges from the systematic manipulation of our stimulus parameters. Implementing and parameterizing such a model in a comparable, biologically grounded framework is a substantial undertaking that lies beyond the scope of the current study. Therefore, our goal here was not to claim exclusivity for synchrony-based mechanisms, but rather to re-evaluate their plausibility by showing that features often seen as limitations (stimulus dependence and frequency heterogeneity) are, in fact, essential characteristics of the TWCO framework that can predict complex behavioral outcomes.

We would also like to clarify that our stimulus features were derived from theory rather than psychophysical literature. Starting from the principles of TWCO, we mapped frequency detuning and coupling strength onto known anatomical and physiological properties of early visual cortex, and only then derived the corresponding stimulus manipulations (contrast heterogeneity and grid coarseness). Demonstrating that these features predict behavior is therefore not trivial but constitutes a first empirical confirmation that the core TWCO variables match perception.

Apart from adding analyses of additional rate-based readouts of our model, we also refined our discussion of the relationship between these and a synchrony-based mechanism.

**Reviewer #2 (Public review):**
The authors aimed to investigate whether gamma synchrony serves a functional role in figure-ground perception. They specifically sought to test whether the stimulus-dependence of gamma synchrony, often considered a limitation, actually facilitates perceptual grouping. Using the theory of weakly coupled oscillators (TWCO), they developed a framework wherein synchronization depends on both frequency detuning (related to contrast heterogeneity) and coupling strength (related to proximity between visual elements). Through psychophysical experiments with texture discrimination tasks and computational modeling, they tested whether human performance follows patterns predicted by TWCO and whether perceptual learning enhances synchrony-based grouping.We thank the reviewer for their thoughtful and constructive review. We believe the comments have served to improve our work.Strengths:(1) The theoretical framework connecting TWCO to visual perception is innovative and well-articulated, providing a potential mechanistic explanation for how gamma synchrony might contribute to both feature binding and separation.(2) The methodology combines psychophysical measurements with computational modeling, with a solid quantitative agreement between model predictions and human performance.(3) In particular, the demonstration that coupling strengths can be modified through experience is remarkable and suggests gamma synchrony could be an adaptable mechanism that improves with visual learning.(4) The cross-validation approach, wherein model parameters derived from macaque neurophysiology successfully predict human performance, strengthens the biological plausibility of the framework.Weaknesses:(1) The highly controlled stimuli are far removed from natural scenes, raising questions about generalisability. But, of course, control (almost) excludes ecological validity. The study does not address the challenges of natural vision or leverage the rich statistical structure afforded by natural scenes.

We agree with the reviewer that the insights of the present study are limited to texture stimuli and have made adjustments in the Discussion (final two paragraphs) to avoid claiming generalizability to natural stimuli. We have also adjusted the title to specifically limit our results to texture stimuli. To establish the principles of TWCO, we needed tight control over the stimulus, but are intrigued by the idea to investigate natural scenes. We have added to our Discussion (paragraph 9) that future should evaluate to what extent the principles we investigate here apply to natural scenes. Synchrony-based mechanisms have been successfully used for image segmentation tasks in machine vision, showing that the proposed mechanism can in principle work for natural scenes.

(2) The experimental design appears primarily confirmatory rather than attempting to challenge the TWCO framework or test boundary conditions where it might fail.

We thank the reviewer for this important point. Our primary motivation was to address the neurophysiological properties of gamma synchrony that have been suggested to severely challenge the binding by synchrony mechanism. Particularly the strong dependence of gamma oscillations and synchrony on stimulus features. Our goal was to show that from the perspective of TWCO, these challenges become expected components of the mechanism. In essence, we wanted to promote a conceptual shift that converts what pushes a theory to its limit into something that is actually its central tenet. To facilitate this shift, we designed the experiment to directly test this core tenet.

While our approach was designed to test a central prediction of TWCO rather than explicitly challenge its boundaries, we respectfully argue that it was far from a simple confirmatory experiment. The design incorporated high-risk elements that provided considerable room for both the theory and our model to fail. First, the core prediction itself was non-obvious and highly specific. We did not simply test whether contrast heterogeneity and grid coarseness affect perception. We tested the stronger hypothesis that they would reflect a specific, interactive trade-off (the behavioral Arnold tongue) as specified by TWCO. Second, our modeling approach was deliberately constrained to provide a further stringent test. We did not post-hoc optimize the model's key parameters to fit our behavioral data. Instead, we fixed them a priori based on independent neurophysiological data from macaques. This was a high-risk choice, as a mismatch between a priori model predictions and the human data would have seriously challenged the framework's generalizability.

We agree that future research should further challenge TWCO. For instance, by using stimuli that require segregating several objects simultaneously or objects that cover more extensive regions of the visual field.

(3) Alternative explanations for the observed behavioral effects are not thoroughly explored. While the model provides a good fit to the data, this does not conclusively prove that gamma synchrony is the actual mechanism underlying the observed effects.

We agree that our results do not conclusively show that gamma synchrony is the actual mechanism underlying figure-ground segregation. We admit that the original phrasing used throughout the manuscript was too strong and gave the impression that we wanted to establish exactly that. However, the goal of our work was only to reinvigorate gamma synchrony as a potential contender by showing that features often cited as limitations of synchrony-based binding may in fact be essential properties of the mechanism. We have revised the title and made adjustments throughout the manuscript to better reflect this more moderate goal.

Additionally, we added tests of alternatives (Results, paragraphs 7–9) to clarify the unique explanatory value of the synchrony mechanism. To that end, we derived two alternative rate-based readouts from the same V1 simulations of our model. First, we extracted average firing rates inside the figure. Second, we computed the difference between average firing rates inside the figure and average firing rates in the background (rate difference). We analyzed each individually as predictors of behavior and performed a model comparison between these two and synchrony based on out-of-sample predictions. While the rate difference (but not average firing) showed meaningful associations with performance when considered alone, the synchrony readout had a larger effect size and was favored by the model comparison.

(4) Direct neurophysiological evidence linking the observed behavioral effects to gamma synchrony in humans is absent, creating a gap between the model and the neural mechanism.

We agree that the model only provides a how-possibly account linking stimulus features to performance. Showing that the brain actually relies on this mechanism would require showing that cortical synchrony mediates the effect of stimulus features on behavior beyond firing rates. Collecting such data would constitute a major effort that would go beyond the scope of this study. We acknowledge the need for electrophysiological data and the mediation analysis in the updated Discussion.

Achievement of Aims and Support for Conclusions:The authors largely achieved their primary aim of demonstrating that human figure-ground perception follows patterns predicted by TWCO principles. Their psychophysical results reveal a behavioral "Arnold tongue" that matches the synchronization patterns predicted by their model, and their learning experiment shows that perceptual improvements correlate with predicted increases in synchrony.The evidence supports their conclusion that gamma synchrony could serve as a viable neural grouping mechanism for figure-ground segregation. However, the conclusion that "stimulus-dependence of gamma synchrony is adaptable to the statistics of visual experiences" is only partially supported, as the study uses highly controlled artificial stimuli rather than naturalistic visual statistics, or shows a sensitivity to the structure of experience.Likely Impact and Utility:This work offers a fresh perspective on the functional role of gamma oscillations in visual perception. The integration of TWCO with perceptual learning provides a novel theoretical framework that could influence future research on neural synchrony.The computational model, with parameters derived from neurophysiological data, offers a useful tool for predicting perceptual performance based on synchronization principles. This approach might be extended to study other perceptual phenomena and could inspire designs for artificial vision systems.The learning component of the study may have a particular impact, as it suggests a mechanism by which perceptual expertise develops through modified coupling between neural assemblies. This could influence thinking about perceptual learning more broadly, but also raises questions about the underlying mechanism that the paper does not address.Additional Context:Historically, the functional significance of gamma oscillations has been debated, with early theories of temporal binding giving way to skepticism based on gamma's stimulus-dependence. This study reframes this debate by suggesting that stimulus-dependence is exactly what makes gamma useful for perceptual grouping.The successful combination of computational neuroscience and psychophysics is a significant strength of this study.The field would benefit from future work extending (if possible) these findings to more naturalistic stimuli and directly measuring neural activity during perceptual tasks. Additionally, studies comparing predictions from synchrony-based models against alternative mechanisms would help establish the specificity of the proposed framework.
**Recommendations for the authors:**

**Reviewing Editor Comments:**
In a joint discussion to integrate the peer reviews and agree on the eLife recommendations, both reviewers agreed that the work is valuable, but they were on the fence about whether the strength of evidence was incomplete or solid, eventually settling on incomplete. The reviewers make several recommendations for improving these ratings, which I (Reviewing Editor) have organised into 3 points below, with point 1 of particular importance. Underneath the summary, please see the individual recommendations of the reviewers.(1) Strengthen evidence for the unique role of gamma synchrony in explaining the data, and ensuring claims are directly supported by relevant data:Reviewers 2 and 3 both note the lack of direct evidence for gamma involvement, and reviewer 2 observes that the fit with behaviour may trivially be explained by a relationship between contrast heterogeneity and grid coarseness without need for oscillation. The reviewers felt that the approach of fitting the model to human data could be strengthened to help address this issue - and they offer various solutions, e.g., more principled a-priori criteria around good vs bad fit of the model to both main task and training data, and comparison to alternative binding models (Reviewer 2), identifying and testing boundary conditions of the model (Reviewer 3). There is also the possibility of collecting direct human neurophysiological evidence linking the behavioural data to neural mechanisms. Our discussion also highlighted the need to weaken claims (including in the title) where links are not directly demonstrated by methods from the present study, e.g., resting on indirect comparisons to primate literature.

We agree with the editor and reviewers that this was a critical point. To address it, we have made several major revisions.

As suggested, we have weakened claims where the links are not directly demonstrated by our data. The title has been revised to be more specific, and we have carefully edited the abstract, introduction, and discussion to distinguish between our model's predictions and direct neurophysiological evidence.

To address the concern that our model's fit might be trivially explained by visual features, we have performed a new analysis comparing the synchrony-based readout to two alternative rate-based readouts from the same V1 simulations. This new comparison shows that the synchrony readout provides a superior out-of-sample prediction of human behavior.

While a full implementation of a competing theory like "Binding by Enhanced Firing" would be a valuable next step, we note that parameterizing such a model in a comparably grounded framework is a substantial undertaking beyond the scope of the present study. Our new analysis provides an important first step in this direction.

(2) Make explicit and address the limitations of the stimuli:Include that the model is not extracting the figure from the background, and the controlled stimuli may limit generalizability.

To address the concern that our model was not performing true figure-ground extraction, we performed a new set of simulations that included both the figure and the immediate background. The results confirm that synchrony dynamics within the figure region are not affected by the presence of the background. We added these validation results as supplementary materials. We have additionally made the modeling choice and its justification more explicit in the Results and Methods sections.

We have revised the Discussion to be more explicit about the limitations of using highly controlled texture stimuli. We now clearly state that our findings are specific to this context and that further research is required to determine if these principles generalize to the segregation of objects in natural scenes.

(3) Some clarifications to make more accessible:Include the figure explaining the framework (Reviewers 1&2), and also the model details (Reviewer 2).

We have revised Figure 1 and its caption to more clearly illustrate the links from TWCO principles to their neural implementation in V1 and the resulting behavioral predictions.

We have expanded the Methods section to provide a more detailed and accessible description of the model's construction. We now clarify precisely how the oscillator grid was defined in visual space, how eccentricity-dependent receptive field sizes were implemented, and how these were mapped onto a retinotopic cortical surface to determine coupling strengths.

**Reviewer #1 (Recommendations for the authors):**
(A) Major concerns:(1) My main concern:My main concern is the repeated claims that the observed findings can be attributed to gamma synchrony in the early visual cortex. I find this claim misleading as the authors do not report any electrophysiological data that directly supports such claims. As stated in my public review, I feel that the authors should be clear about direct evidence versus more abstract inferences based on the literature.In particular, I recommend changing claims about "gamma synchrony" to "Binding by Synchrony" That being said, the authors can outline that the model was built under the assumption that this synchrony is mediated by gamma in early visual cortex, but I don't think it should be part of their main conclusions.

We appreciate that TWCO’s general principles are frequency-agnostic and can be viewed as binding by synchrony in a broad sense. Our work, however, specifically instantiates these principles in V1 gamma: the model reflects TWCO dynamics together with V1 anatomy/physiology and the well-established contrast–frequency relationship in the gamma range (which, to our knowledge, has not been demonstrated with comparable specificity for other bands). In that sense, it is a gamma oscillator model of V1, rather than a generic BBS instantiation. Moreover, stimulus dependencies often cited as challenges to BBS have been used in particular to argue against gamma; showing that these very dependencies are integral to the TWCO mechanism is central to our contribution, and we therefore keep our conclusions focused on the gamma-specific instantiation tested here.

(2) Mediation of the observed effects by the visual features of the figure:The authors motivate the hypothesis that BBS predicts that the perception of texture-defined objects depends on the density of texture elements and their contrast heterogeneity. This hypothesis seems trivial as those are the features that distinguish figure from ground. I think it would be important to clarify how this hypothesis is unique to BBS and not explained by competing theories, such as Binding by Enhanced Firing (Roelfsema, 2023). The authors should be clear about what part of the hypothesis is not trivial based on the task and clearly attributable to oscillators and synchrony.

Our stimulus features were derived from theory rather than psychophysical literature. Starting from the principles of TWCO, we mapped frequency detuning and coupling strength onto known anatomical and physiological properties of early visual cortex, and only then derived the corresponding stimulus manipulations (contrast heterogeneity and grid coarseness). We agree that grid coarseness (element distance) is an established facilitator of figure–ground perception. By contrast, contrast heterogeneity (feature variance) is less commonly emphasized as a figure–ground cue, compared to mean-based cues, but follows directly from TWCO’s frequency detuning. Importantly, mean contrast and luminance were matched exactly between figure and background in our stimuli. Demonstrating that contrast heterogeneity and grid coarseness not only independently affect figure-ground perception, but reflect a trade-off where higher heterogeneity needs to counteracted by reduced grid coarseness in the way TWCO specifies is therefore non-obvious and provides an initial empirical indication that the core TWCO variables might shape perception. We also agree that alternative models would further clarify the unique explanatory value of synchrony. In the revised manuscript, we compare rate-based readouts (mean figure rate; figure–background rate difference) with the synchrony readout from the same simulations. Rate difference indeed constitutes a predictor of performance, but the synchrony readout showed a larger effect and was preferred by out-of-sample model comparison.

Using a linear model, the authors assess the relationship between discrimination accuracy and synchrony. Did the authors also include the factors grid coarseness and contrast heterogeneity in this model? Again, as both the task performance (as shown by the GEE analysis) and oscillatory synchrony depend on these features, the relationship between model and behavioral performance will be mediated by the visual features.

Thank you for raising this. In our framework, detuning (via contrast heterogeneity) and coupling (via grid coarseness) are the inputs, synchrony is the proposed mechanistic mediator, and behavior is the output. Because synchrony in our model is a (near-)deterministic function of the manipulated features under fixed parameters, a joint features+synchrony regression is statistically ill-posed (perfect multicollinearity up to numerical error) and cannot add information. A proper mediation test would require trial-wise neural measurements of synchrony in the same task, which we do not have and acknowledge as a limitation in the Discussion. Accordingly, we show that both the features themselves (reflecting TWCO principles) and model-derived synchrony (realizing the proposed pathway) account for behavior.

We agree this does not establish a unique contribution of synchrony. To probe alternatives, we added rate-based readouts and a model comparison to the revised manuscript. These additional analyses indicate that synchrony outperforms simple rate-based mappings. We do not claim this rules out more sophisticated rate-based mechanisms. Our aim is to demonstrate that synchrony is a viable, behaviorally informative readout for downstream processing. We do not assert it is the only mechanism the brain uses. Synchrony had been discounted due to its stimulus dependence; our results are intended to rule it back in. We have made changes throughout the manuscript to better reflect this more modest aim.

(3) Goodness of fit measures are not established a prior:I have described this concern in my public review. It is hard to assess what the authors would have interpreted as a good or a bad fit, especially without accounting for the confound in the relationship between oscillator synchrony and behavior. Similarly, when assessing the similarity between the behavioral and dynamic Arnold Tongues across different coupling parameters, the authors found that the chosen parameters (based on macaque data) were not optimal. They offer the explanation that the human cortex has a lower coupling decay than the macaque cortex, and the similarity is higher for lower values of coupling decay. While this explanation is not entirely implausible, it is unclear where an oscillator model with human values would be in the presented plot, as the authors didn't estimate those values from the human studies. Moreover, the task used in the Lowet et al., 2017 paper is very different from the task presented here, which could also account for differences. Overall, the explanation appears hand-wavy considering the lack of empirically defined goodness of fit measures.

Thank you for these concerns.

We did indeed not provide a priori thresholds for what would be considered good fit. Instead, we used two complementary benchmarks; namely noise ceilings and parameter exploration. The former provides an upper bound on what any model (not just ours but based on completely different mechanisms) could achieve given our data. The parameter sweep provides an indication how well our concrete model can maximally fit the data and how bad it can be based on possible parameters. These benchmarks are more informative than a fixed a-priori cutoff, which would depend on unknown noise and inter-subject variability. Both the noise ceiling and the parameter exploration indicate that our model, using a priori fixed parameters, performs well. Additionally, we redid all our statistical analyses after z-normalizing every predictor to provide easier interpretation of effect sizes.

Regarding the reason that key model parameters were not optimal, we believe our interpretation to be plausible. We agree that we currently do not have data to estimate the exact human decay factor and hence cannot establish how much model fit would be affected. However, the parameter exploration in Figure 3 shows that small to modest reductions in decay would improve model fit. We discuss this now in the revised manuscript.

The reviewer’s suggestion is intriguing. While Lowet et al. (2017) used a different task, the parameters we took from their work (decay rate and maximum coupling) are intended to reflect anatomical properties and thus should not be task-dependent. That said, Lowet et al. ‘s data carry uncertainty, so our estimates may not be exact; we note this explicitly in the revised Discussion. Whether a different task would have yielded better parameter estimates is difficult to determine, but we considered Lowet’s paradigm appropriate because it was designed to target the same V1 anatomical and physiological properties that map onto TWCO.

I have concerns about a similar confound in the training effects. If I'm not mistaken, the Hebbian Learning rule encourages synchronization between the oscillators in the grid. As such, it causes synchronization to increase over several simulations. Clearly, the task performance of the participants also improves over the sessions. Again, an empirical threshold would be required to assess whether the similarity in learning between model and performance goes beyond what is expected based on learning alone. How much of these effects can be attributed to the model being oscillatory?

The reviewer is correct that, in our framework, learning operates via changes in coupling that increase synchrony. Enhanced synchrony is the proposed (and in our model also the actual) pathway by which learning impacts behavior. We agree that learning could, in principle, act through pathways other than synchrony. Demonstrating this would not be achieved by a mediation analysis here, because that requires independent, trial-level neural measurements of the candidate pathways (synchrony and alternatives). In the absence of such data, the appropriate approach would be model comparison between competing mechanistic readouts. We have added such a model comparison for a synchrony readout versus two rate-based readouts derived from the same simulations for the first session; i.e., focusing on the pathway from stimulus features to behavior. However, a similar model comparison is not possible for learning. As we show in the supplementary materials, rate-based readouts of our V1 model are not at all affected by coupling strength. As such, they are insensitive to changes in coupling and are thus not viable as alternative mechanisms to explain performance changes due to learning. A fair test of rate-based alternatives would require building a detailed rate-based figure–ground segregation model that predicts session-wise changes. We agree that this is an important next step but it is also substantial undertaking beyond the scope of the present study.

(4) Similarly, for the comparison of the Arnold Tongue in the transfer session and the early session:In the first part of the Results section, it says: "Our model rests on the assumption that learning-induced structural changes in early visual cortex are specific to the retinotopic locations of the trained stimuli. We evaluated whether this assumption holds for our human participants using the transfer session following the main training period. [...] If learning is indeed local, participants' performance in the transfer session should resemble that of early training sessions, indicating a reset in performance for the new retinal location."The authors find that a model fit to session 3 explains the data in the transfer session best and consider this as evidence for the above-stated expectation. Again, it is unclear where the cutoff would have been for a session to be declared as early or late. For instance, had the participants only performed 4 sessions, would the performance be best explained by session 3 or session 1?A high number of statistical tests are used, which, firstly, need to be corrected for multiple comparisons (did the authors do this?). Secondly, I feel that the regression models could be improved. For instance, the authors fit one model per session and then assess how well each model explains the variance in the transfer session. I think the authors might want to opt for one model with the regressors contrast heterogeneity, grid coarseness, and session (and their interaction). Using this approach, the authors would still be able to assess which session predicts the data best. Similarly, interindividual variability could be accounted for by adding participant-specific random effects to the model (and using a mixed model), instead of fitting individual models per participant.

We agree the “early vs late” cutoff was underspecified. In the revision, we predefine Session 2 as the early-learning reference, excluding Session 1 to avoid familiarization/response–mapping effects. We then fit a single Bayesian hierarchical model with contrast heterogeneity, grid coarseness, and session, plus a transfer indicator, and participant-level random effects. This allows us to place the transfer session on the same scale as training and to test (a) whether the transfer session precedes the state in session 2 via the posterior contrast P(βtransfer<βSess2) and (b) whether it is indistinguishable from the state in session two using an equivalence test derived from the fitted model. We find that the transfer session is equivalent to session 2. We added this updated analysis of the transfer session in the Results (paragraph 15).

In response to the suggestion to use a hierarchical regression model for analyzing the transfer session, we have decided to use such a model for all our analyses in a Bayesian framework. In this Bayesian framework, inference is based on the joint posterior (credible intervals/equivalence) of all predictors in a model and additional post-hoc multiplicity corrections are not required.

(5) Questions regarding the model:What does it mean that the grid was "defined in visual space"? How biologically plausible with regard to the retinotopy and organization of the oscillators do the authors claim the model to be?

We are happy to clarify this point. We have a total of 400 oscillators reflecting neural assemblies in V1. We start by defining a regular, 20x20, grid of the receptive field (RF) centers of these oscillators inside the figure region. Each oscillator is then also assigned a RF size based on the eccentricity of its RF center. We use the threshold-linear relationship between RF eccentricity and RF size reported in [1] to assign RF sizes. Each oscillator thus has an individual, eccentricity-dependent, RF size.

For the coupling between oscillators, we need to know their cortical distances. We obtain these by first determining the cortical location of each oscillator through a complex-logarithmic topographic mapping of neuronal receptive field coordinates onto the cortical surface [2,3]. For this mapping, we use human parameter values estimated by [4]. From these cortical locations, we then compute pairwise Euclidean distances.

The model thus captures realistic retinotopy, eccentricity-dependent RF sizes, and distance-dependent coupling on the cortical surface. We have adjusted our Methods to make these steps clearer.

(1) Freeman, J., & Simoncelli, E. P. (2011). Metamers of the ventral stream. Nature neuroscience, 14(9), 1195-1201.

(2) Balasubramanian, M., & Schwartz, E. L. (2002). The isomap algorithm and topological stability. Science, 295(5552), 7. https://doi.org/10.1126/science.1066234

(3) Schwartz, E. L. (1980). Computational anatomy and functional architecture of striate cortex: a spatial mapping approach to perceptual coding. Vision Research, 20(8), 645–669. http://www.sciencedirect.com/science/article/pii/0042698980900905

(4) Polimeni, J. R., Hinds, O. P., Balasubramanian, M., van der Kouwe, A. J. W., Wald, L. L., Dale, A. M., & Schwartz, E. L. (2005). Two-dimensional mathematical structure of the human visuotopic map complex in V1, V2, and V3 measured via fMRI at 3 and 7 Tesla. Journal of Vision, 5(8), 898. https://doi.org/10.1167/5.8.898

Similarly, do the authors claim that each gabor annuli stimulates a single receptive field in V1?

We hope that with the additional explanation above, it is clearer that there is not a one-to-one mapping. Each oscillator samples the local image by pooling over all Gabor annuli that overlap its receptive field (partially or fully) and computes the average contrast within its RF. Conversely, a single annulus typically overlaps multiple RFs and contributes to each in proportion to the overlap.

I am unsure how the oscillators were organized, if not retinotopically. How is the retinotopic input fed into the non-retinotopically arranged oscillators?

We hope that with the additional explanation above, it is clearer that the network is strictly retinotopic.

The frequency of each oscillator changes according to ω=2πv with ν=25+0.25C. How were the values for the linear regression in v chosen? Reference?

The slope and intercept parameters for this equation were first reported in [5]. We added the reference to the Methods.

(5) Lowet, E., Roberts, M., Hadjipapas, A., Peter, A., van der Eerden, J., & De Weerd, P. (2015). Input-dependent frequency modulation of cortical gamma oscillations shapes spatial synchronization and enables phase coding. PLoS computational biology, 11(2), e1004072.

(6) Hebbian Learning Rule:I am confused about how the effective learning rate E = ∈t is calculated. It is said that it is estimated based on the similarity between the second experimental session and the distribution of synchrony after letting the model learn. How can the model learn without knowing epsilon and t?

We agree with the reviewer that our procedure to estimate the effective learning rate requires further clarification. We performed a nested grid search. Essentially, we let the model learn between session 1 and 2 with each of 25 candidate effective learning rates and evaluate how well each of them allow the model to fit performance in session 2. We then select the best effective learning rate and create a new, smaller, grid around this value and repeat that procedure. In total we perform 5 nested grids to arrive at the final effective learning rate. We expanded the explanation in the Methods.

(B) Minor concerns:(1) Small N: 2/3 of the studies that were cited to justify the small sample were notably different from the current experiment, i.e., Intoy 2020 is an eye movement task, Lange 2020 is a memory task (Tesileanu 2020 is more similar). I think a power analysis would be great to support, as the sample size seems quite low

Our study uses a within-subject design with ~750 trials per session (≈6,000 total) per participant, analyzed with a hierarchical model that pools information across trials and participants. To assess adequacy, we ran a simulation-based design analysis using the fitted hierarchical model (i.e., post hoc, based on the observed variance components). This analysis indicated a detection probability >90% for all key effects. We now report the results of this design analysis in the (Supplementary Table 1) and note this in the Results (paragraph 1).

Regarding the literature context, we agree the cited studies are not identical to ours; we referenced them to illustrate a common practice (small N with many trials) when targeting low-level, early-visual mechanisms. Intoy (pattern/contrast sensitivity) and Lange (perceptual learning in early vision) share that focus, while Tesileanu is methodologically closest.

(2) Figure 1 could be more informative and better described in the text. The authors often don't refer to the panels in Figure 1. Maybe it would help to swap a and b to describe the Arnold tongue first? It might also be a good idea to add the coupling strength and frequency detuning axes

We have swapped panels a and b and now refer to each panel in the main text to enhance clarity.

(3) Values of rho (distance - is this degrees visual angle)? Do the authors assume that the size of the stimuli corresponds to receptive fields in V1? If so, how is this justified?

The center-to-center distance between any pair of neighboring annuli is indeed expressed in degrees of visual angle. Rho is a scaling factor for this distance. With rho=1, the center-to-center distance corresponds to the diameter of the annuli; i.e., they touch but do not overlap each other. We do not assume any relation between the size of receptive fields and the size of the annuli. Receptive field sizes in our model are purely determined by their eccentricity and each oscillator can have several annuli within its receptive field while each annulus can fall within several overlapping receptive fields of different oscillators. We believe that the schematic illustration in Figure 1 might have given the impression that each oscillator sees exactly one annulus and added a note that this is not the case and merely an oversimplification to illustrate the relationship between contrast and intrinsic frequency.

(4) Some equations are embedded in the text, and some are not. It might be easier to find the respective equation if they all have an index. For instance, the authors mention the psychometric function that relates model synchrony and performance in the results section. It would be easier to find if it had an index that the authors could refer to.

We moved this equation as well as the contrast intrinsic frequency mapping from inline to displayed and numbered them.

(5) Is there a reference for "Our model rests on the assumption that learning-induced structural changes in early visual cortex are specific to the retinotopic locations of the trained stimuli"? (If so, it should be cited.)

We added references supporting this assumption.

(6) Figure 2b: colorbar missing label.

We added the label.

**Reviewer #2 (Recommendations for the authors):**
Cool work!(1) The reader would benefit from (a single) comprehensive figure that visually explains the entire conceptual framework-from TWCO principles to neural implementation to behavioural predictions-accessible to readers without specialised knowledge of oscillatory dynamics. This will give the paper a greater impact.

We have adjusted Figure 1 in accordance with suggestions made by reviewer 1 and added further explanations to the caption and the Introduction to enhance clarity on how the principles of TWCO relate to neural implementation.

(2) I think this paper would benefit from the audience eLife provides, but the paper could move closer to the audience.(3) Pride comes before the fall, but I am not the most uninformed reader, and it took me some effort to process everything.

Thank you, we took this to heart. In the Introduction, we now state more explicitly how each variable is operationalized and how these map onto TWCO with improved reference to relevant panels in the schematic figure. We agree the framework is conceptually dense. TWCO principles reach the stimuli through specific V1 anatomy and physiology, so there are several links to keep in mind. Our goal with the revised introduction and figure is to make those links better visible.

(4) You could consider discussing potential implications for understanding perceptual disorders characterized by altered neural synchrony (e.g., schizophrenia, autism) and how your learning paradigm might inform perceptual training interventions.

Thank you for this suggestion. We have added that TWCO might provide a new lens to study perceptual disorders to the Discussion. We provide a concrete example of the relation between grouping, gamma synchrony (in light of TWCO) and lateral connectivity in schizophrenia

(5) I think this paper has real strength, but rather than dispersing limitations throughout the discussion, create a dedicated section that systematically addresses ecological validity, alternative explanations, and generalisability concerns. This will also preempt criticism.

We appreciate the suggestion. Our preference is to discuss limitations in context, next to the specific results they qualify, so readers see why each limitation matters and how it affects interpretation. Nevertheless, paragraph 7 on page 20 summarizes most limitations in a single paragraph.